# DIFFERENTIALLY PRIVATE LEARNERS FOR HETEROGENEOUS TREATMENT EFFECTS

**Maresa Schröder, Valentyn Melnychuk & Stefan Feuerriegel**
LMU Munich
Munich Center for Machine Learning (MCML)
{maresa.schroeder,melnychuk,feuerriegel}@lmu.de

## ABSTRACT

Patient data is widely used to estimate heterogeneous treatment effects and thus understand the effectiveness and safety of drugs. Yet, patient data includes highly sensitive information that must be kept private. In this work, we aim to estimate the conditional average treatment effect (CATE) from observational data under differential privacy. Specifically, we present DP-CATE, a novel framework for CATE estimation that is *Neyman-orthogonal* and further ensures *differential privacy* of the estimates. Our framework is highly general: it applies to any two-stage CATE meta-learner with a Neyman-orthogonal loss function, and any machine learning model can be used for nuisance estimation. We further provide an extension of our DP-CATE, where we employ RKHS regression to release the complete CATE function while ensuring differential privacy. We demonstrate our DP-CATE across various experiments using synthetic and real-world datasets. To the best of our knowledge, we are the first to provide a framework for CATE estimation that is Neyman-orthogonal and differentially private.

## 1 INTRODUCTION

Machine learning (ML) is increasingly used for estimating treatment effects from observational data (e.g., Baiardi & Naghi, 2024; Braun & Schwartz, 2024; Ellickson et al., 2023; Feuerriegel et al., 2024). Yet, this involves sensitive information about individuals, and, hence, methods are often needed to ensure privacy.

**Motivating example:** *Electronic health records (EHRs) are commonly used to estimate treatment effects and thus to personalize care. Yet, EHRs capture highly sensitive data about patients (Brothers & Rothstein, 2015). Hence, many regulations, such as the US Health Insurance Portability and Accountability Act (HIPAA), mandate strong privacy guarantees for ML in medicine.*

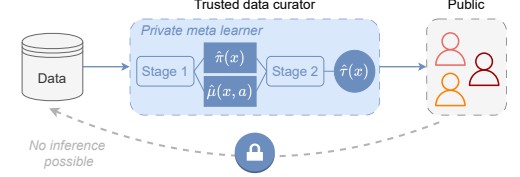

Figure 1: **Setting: CATE estimation under DP.** Only the trusted data curator can access the data, while published CATE estimates do not allow private information about individuals to be inferred.

To ensure the privacy of information contained in the training data of ML models, multiple *privacy mechanisms* have been introduced. Arguably, the most common mechanism is *differential privacy* (DP) (Dwork, 2006; Dwork & Lei, 2009). DP builds upon the idea of injecting noise into algorithms so that sufficient information about the complete population in a dataset is kept while safeguarding sensitive information about individuals. Importantly, DP enjoys stringent theoretical guarantees and is widely used across different fields (e.g., Abadi et al., 2016; Bassily et al., 2014; Wang et al., 2019).

However, methods for treatment effect estimation under DP are scarce. Existing work has primarily focused on the *average treatment effect* (ATE) (e.g., Lee et al., 2019; Ohnishi & Awan, 2023). However, the ATE fails to capture important variations in how different subgroups or individuals respond to treatments. Therefore, many applications such as personalized medicine are interested in the *conditional average treatment effect* (CATE) (e.g., Ballmann, 2015; Feuerriegel et al., 2024).

In this paper, we estimate the CATE from observational data under DP (Fig. 1). Specifically, we propose DP-CATE, an output perturbation mechanism for Neyman-orthogonal CATE estimators that

satisfies DP. Neyman-orthogonal estimators are generally preferred over standard plug-in estimators, as they are less dependent on the estimation errors of nuisance functions (Morzywolek et al., 2023).

Our DP framework is highly flexible and can be combined with all weighted Neyman-orthogonal two-stage CATE learners, such as the R-learner (Nie & Wager, 2020). Further, our framework is model-agnostic and can be used with any ML model as a base learner. To the best of our knowledge, we are the first to provide a framework for Neyman-orthogonal CATE estimation under differential privacy. Our DP-CATE is designed for two use cases relevant to medical practice:

①  *Finitely many queries:* Reporting research findings about medical studies that involve sensitive data requires that finitely many CATE values are estimated, such as treatment effects of a drug for various patient subgroups. In this setting, we treat the different CATE estimates as a potentially high-dimensional vector, for which we derive DP guarantees. Interestingly, we later employ a largely unexplored connection between Neyman-orthogonality and privacy, which allows us to base our DP-CATE on efficient influence functions ($\rightarrow$ our Theorem 1).

②  *Functional query:* Medical researchers may want to have access to the *complete CATE function*. This is relevant when deploying a CATE function in clinical decision support systems where predictions about treatment effects are made for every incoming patient. Hence, this requires querying the CATE function a large number of times, but where the exact number is a priori unknown. In this case, the respective CATE vector would have an infinite dimension, and, as a result, privately releasing the complete CATE function cannot be performed in the same manner as in the first use case. As a remedy, we derive a tailored privacy framework for functional queries, where we make use of tools from functional analysis to calibrate a Gaussian process, which we add to the CATE function estimated through RKHS regression ($\rightarrow$ our Theorem 3).

*Why is it non-trivial to derive privacy mechanisms for CATE estimation?* Common DP strategies include perturbations of either the data, model, or output (e.g., Abadi et al., 2016; Chaudhuri et al., 2011). Yet, a naïve application of such perturbations would naturally violate causal assumptions and/or lead to CATE estimates that are biased. Furthermore, the CATE is an unobservable, functional quantity. However, common privatization mechanisms are only developed for vector-valued quantities. Thus, it is **not** possible to follow the standard procedure of adding calibrated noise to the algorithm. Rather, we have to derive a novel, non-trivial framework that is tailored to our setting.

**Our contributions:**[1]  (1) We propose a novel framework for CATE estimation that is differentially private and Neyman-orthogonal. (2) We extend our framework to privately release both CATE estimates and even the complete CATE function. (3) We demonstrate our proposed framework for differentially private CATE estimation in experiments across various datasets.

## 2    RELATED WORK

We provide a brief overview of the different literature streams relevant to our work, namely, (i) CATE estimation, (ii) differential privacy, and (iii) works that adapt DP to treatment effect estimation.

**CATE estimation:** Popular methods for estimating CATE from observational data are the Neyman-orthogonal meta-learners, such as the DR-learner (Kennedy, 2023a; van der Laan, 2006) and the R-learner (Nie & Wager, 2020). A strength of meta-learners is that these are model-agnostic and can thus be instantiated with arbitrary ML models (e.g., neural networks). Neyman-orthogonal learners (also known as debiased learners) have several additional benefits. (i) The learners achieve quasi-oracle efficiency[2], which is guaranteed due to the first-order insensitivity to errors in the estimation of nuisance functions. As a result, the estimators are asymptotically equivalent to the oracle estimator (= the estimator that has access to the oracle nuisance functions), thereby mitigating the finite-sample bias arising from the misspecification of the nuisance functions (Mackey et al., 2018; Morzywolek et al., 2023). (ii) The DR-learner is further doubly robust: it yields consistent estimation even when either the outcome or the propensity model is not correctly specified. (iii) The R-learner is less sensitive to overlap violations due to its overlap weighting of the loss.

In this paper, we focus on DP for Neyman-orthogonal meta-learners. We derive a framework for the R-learner in the main paper and provide an extension to the DR-learner in Supplement B.

---

[1] The source code is available at our GitHub repository.

[2] Informally, quasi-oracle efficiency means that the target model is learned almost equally well using either the estimated nuisance functions or the ground truth.

**Differential privacy:** DP ensures that the release of aggregated results does not reveal information about individual data samples, typically with strict theoretical results (Dwork, 2006; Dwork & Lei, 2009). As a result, DP has been employed in various fields of machine learning (e.g., Abadi et al., 2016; Bassily et al., 2014; Wang et al., 2019), but typically *outside* of CATE estimation. We discuss different strategies on how DP can be achieved in Supplement A. For example, one strategy is output perturbation, where one adds calibrated random noise to the non-private model prediction prior to its release in order to ensure DP (e.g., Chaudhuri et al., 2011; Zhang et al., 2022).

In our setting, we later adopt output perturbation to CATE estimation. Output perturbation has two clear advantages in our task: (i) it can be applied to *any* ML model after training, which naturally fits the idea of meta-learners from above as model-agnostic approaches; and (ii) it leaves the original objective and the data unchanged. The latter is crucial because changes to the input or the objective (i.e., the estimand) arguably could violate causal assumptions and lead to biased results.

**DP in treatment effect estimation:** The existing literature on DP methods for treatment effect estimation is sparse. We provide an overview in Fig. 2, where we group prior works by the underlying estimand: ● *Average treatment effect (ATE).* Many works focus on privately estimating the ATE (e.g., Javanmard et al., 2024; Lee et al., 2019; Yao et al., 2024). Yet, the ATE is a much simpler causal quantity than the CATE. It makes population-wide estimates and, thus, unlike the CATE, does *not* allow to make individualized predictions about

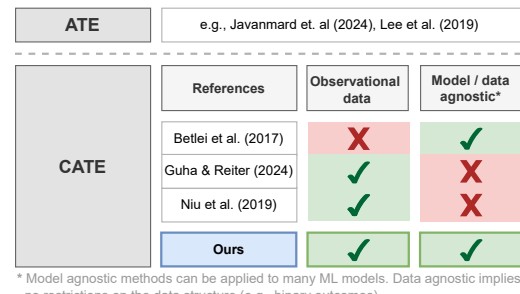

Figure 2: Comparison of relevant literature.

treatment effects. ● *Conditional average treatment effect (CATE).* The few works aimed at DP for CATE estimation have clear *limitations* (Fig. 2): they (i) are either restricted to interventional data from RCTs and thus *not* applicable to observational data (Betlei et al., 2021); (ii) are *only* applicable to binary outcomes (Guha & Reiter, 2024); or (iii) require special private base learners and are thus *not* model-agnostic (Niu et al., 2022). In particular, the latter work (Niu et al., 2022) is restricted to explainable boosting machines. Therefore, it is *not* applicable to other models such as neural networks. In sum, none of the above works provide a method for CATE estimation under DP where both observational data and different types of ML models can be used.

**Research gap:** So far, a DP framework for CATE estimation from observational data with meta-learners is missing. We are thus the first to propose a Neyman-orthogonal framework for CATE estimation that fulfills DP.

## 3 PROBLEM FORMULATION

**Notation:** We write random variables as capital letters $X$ with realizations $x$. We denote the probability distribution over $X$ by $P_X$, where we omit the subscript whenever it is apparent from the context. We denote the probability mass function / density function by $P(x) = P(X = x)$. We rely on the potential outcomes framework (Rubin, 2005) and denote the outcome under intervention $a$ by $Y(a)$. Finally, $\|\mathbf{z}\|_2 = \sqrt{z_1^2 + \cdots + z_d^2}$ is an $l_2$ norm for $\mathbf{z} \in \mathcal{Z}^d$; $\|f\|_{L_k} = (\mathbb{E}|f(Z)|^k)^{1/k}$ is an $L_k$ norm; $a \lesssim b$ means there exists $C \geq 0$ such that $a \leq C \cdot b$; and $X_n \in o_P(r_n)$ means $X_n/r_n \xrightarrow{p} 0$.

**Setting:** We consider a dataset $\bar{D} := \{(X_i, A_i, Y_i)\}_{i=1,\ldots,2n}$, consisting of observed confounders $X$ in a bounded domain $\mathcal{X} \subseteq \mathbb{R}^q$, a binary treatment $A \in \{0, 1\}$, and a bounded discrete or continuous outcome $Y \in \mathcal{Y}$, where $Z_i := (X_i, A_i, Y_i) \sim P$ i.i.d., $Z_i \in \mathcal{Z}$. Let $\pi(x) := P(A = 1 \mid X = x)$ define the propensity score and $\mu(x, a) := \mathbb{E}[Y \mid X = x, A = a]$ the outcome function.

**Estimand:** Our objective is to estimate the conditional average treatment effect (CATE) $\tau(x)$ for individuals with covariates $X = x$. We make the standard assumptions for causal treatment effect estimation: positivity, consistency, and unconfoundedness (e.g., Curth & van der Schaar, 2021; Rubin, 2005). [3] Then, the CATE is identifiable as

$$\tau(x) := \mathbb{E}[Y(1) - Y(0) \mid X = x] = \mu(x, 1) - \mu(x, 0). \tag{1}$$

---

[3]We give more details on the standard causal assumptions and CATE estimation in Supplement A.4.

In our work, we aim to estimate $\tau(x)$ by (i) using Neyman-orthogonal meta-learners (Sec. 3.1) while (ii) ensuring differential privacy (Sec. 3.2), which we briefly review in the following.

## 3.1 Neyman-orthogonal meta-learners for CATE estimation

To estimate the CATE from Eq. (1), the common approach is to regress the difference between the potential outcomes (i.e., $Y(1) - Y(0)$) on the covariates $X$. Thus, one considers the population risk for a working model $g \in \mathcal{G} : \mathcal{X} \mapsto \mathbb{R}$ via

$$R_P(g, \eta, \lambda(\pi)) = \mathbb{E}[\lambda(\pi(X))\left((\mu(X, 1) - \mu(X, 0)) - g(X)\right)^2] + \Lambda(g), \qquad (2)$$

where $\eta = (\mu, \pi)$ are nuisance functions, $\lambda(\cdot) > 0$ is a *weight function*, and $\Lambda(g)$ is a regularization term. We denote a population minimizer of $R_P(g, \eta, \lambda(\pi))$ with $g^*(\cdot; \eta) = \arg\min_{g \in \mathcal{G}} R_P(g, \eta, \lambda(\pi))$ (Hirano et al., 2003; Morzywolek et al., 2023). However, $R_P$ cannot be directly estimated and subsequently minimized given the data $\bar{D}$, as it depends on the unknown nuisance functions $\pi$ and $\mu$. One could employ the estimated nuisance functions (i.e., $\hat{\pi}_{\bar{D}}$ and $\hat{\mu}_{\bar{D}}$) here, but then, their estimation errors propagate into the minimization of $R_P$.

A popular approach to circumvent the above problem is to use Neyman-orthogonal meta-learners. Formally, such meta-learners operate in two stages (e.g., Kennedy, 2023b; Nie & Wager, 2020). Let the dataset $\bar{D}$ be a disjoint union of the two subsets $\tilde{D}$ and $D$ of size $n$, i.e., $\bar{D} = \tilde{D} \cup D$. In the first stage, the meta-learners estimate *nuisance functions* $\hat{\eta}_{\tilde{D}} = (\hat{\pi}_{\tilde{D}}, \hat{\mu}_{\tilde{D}})$ on dataset $\tilde{D}$, and, in the second stage, we minimize the adapted Neyman-orthogonal population risk function

$$R_P(g, \eta, \lambda(\pi)) = \mathbb{E}\left[\rho(A, \pi(X))\left(\phi(Z, \eta, \lambda(\pi(X))) - g(X)\right)^2\right] + \Lambda(g) \qquad (3)$$

with
$$\rho(a, \pi) := (a - \pi(x))\lambda^{'}(\pi(x)) + \lambda(\pi(x)) \quad \text{and} \qquad (4)$$

$$\phi(z, \eta, \lambda(\pi)) := \frac{\lambda(\pi(x))}{\rho(a, \pi(x))} \frac{a - \pi(x)}{\pi(x)\,(1 - \pi(x))}(y - \mu(x, a)) + \mu(x, 1) - \mu(x, 0) \quad (5)$$

on dataset $D$, where $\hat{\eta}_{\tilde{D}}$ is used in place of $\eta$ (Morzywolek et al., 2023). The use of this loss function allows for quasi-oracle efficiency of the final estimator. In the following, we denote the estimated population risk $R_P$ via a loss $R_D$ that is dependent on the data $D$, namely

$$R_D(g, \eta, \lambda(\pi)) = \frac{1}{n}\sum_{i=1}^{n} \rho(A_i, \pi(X_i))\left(\phi(Z_i, \eta, \lambda(\pi(X_i))) - g(X_i)\right)^2 + \Lambda(g). \qquad (6)$$

Later, we use the R-learner (Nie & Wager, 2020), which is given by $\lambda^{\mathrm{R}}(\pi) = \pi(x)\,(1 - \pi(x))$, due to its theoretical advantages (e.g., Neyman-orthogonality and oracle-efficiency). Importantly, Neyman-orthogonal meta-learners such as the R-learner achieve state-of-the-art performance, and the orthogonality property makes the models less sensitive to the misspecification of the nuisance functions (e.g., Curth & van der Schaar, 2021; Kennedy, 2023b; Melnychuk et al., 2025).

## 3.2 Differential privacy

Differential privacy (DP) ensures that the inclusion or exclusion of data from any individual does not significantly affect the estimated outcome (Dwork, 2006; Dwork & Lei, 2009). Specifically, for a given *privacy budget* $\varepsilon$, DP ensures that the probability density of any outcome $y$ on dataset $D \in \mathcal{Z}^n$ is $\varepsilon$-*indistinguishable* from the probability density of the same outcome $y$ stemming from a neighboring dataset $D^{'} \in \mathcal{Z}^n$ with a probability of at least $1 - \delta$. The datasets $D$ and $D^{'}$ are called *neighbors*, denoted as $D \sim D^{'}$, if their Hamming distance equals one, i.e., $d_{\mathrm{H}}(D, D^{'}) = 1$.

**Definition 1** (Differential privacy (Dwork & Lei, 2009)). *A function $\hat{f}_D(\mathbf{x}) : \mathcal{X}^d \mapsto \mathbb{R}^d$ trained on a dataset $D$ is $(\varepsilon, \delta)$-differentially private if, for all neighboring datasets $D$, $D^{'} \in \mathcal{Z}^n$ and all measurable $S \subseteq \mathbb{R}^d$, it holds that*

$$P(\hat{f}_D(\mathbf{x}) \in S) \leq \exp(\varepsilon) \cdot P(\hat{f}_{D'}(\mathbf{x}) \in S) + \delta \qquad \text{for all } \mathbf{x} \in \mathcal{X}^d. \qquad (7)$$

One common strategy to ensure DP is *output perturbation* (Chaudhuri et al., 2011; Zhang et al., 2022), which we later tailor to CATE estimation as part of our framework. Intuitively, one perturbs the prediction in a way that the predictions resulting from two neighboring databases cannot be differentiated. It has been shown (e.g., Dwork & Roth, 2014) that adding appropriately calibrated zero-centered noise (e.g., Gaussian noise) to the prediction is sufficient to ensure differential privacy for *traditional*, supervised machine learning tasks (but not for CATE estimation, as we discuss later). This is stated in the following *Gaussian noise privacy mechanism.*

**Definition 2** (Gaussian noise privacy mechanism (Dwork & Roth, 2014)). *Let $\hat{f}_D(\mathbf{x}) : \mathcal{X}^d \mapsto \mathbb{R}^d$ be a function on dataset $D$ with $l_2$-sensitivity $\Delta_2(\hat{f}) = \sup_{D \sim D', \mathbf{x} \in \mathcal{X}^d} \|\hat{f}_D(\mathbf{x}) - \hat{f}_{D'}(\mathbf{x})\|_2$ and $\mathbf{U} \sim \mathcal{N}(0, \sigma \boldsymbol{I}_d)$ for $\sigma \geq \frac{1}{\varepsilon} \sqrt{2 \ln(1.25/\delta)} \, \Delta_2(\hat{f})$. Then, the output perturbation mechanism returns $\mathcal{M} : \hat{f}_{\mathrm{DP}}(\mathbf{x}) = \hat{f}_D(\mathbf{x}) + \mathbf{U}$ that preserves $(\varepsilon, \delta)$-differential privacy.*

Definition 2 describes how to ensure DP for a given prediction. However, this requires estimating the training sensitivity $\Delta_2(\hat{f})$ of the employed model $\hat{f}$, which, for general function classes such as neural networks, is infeasible. Hence, this motivates our custom framework later.

## 3.3 PROBLEM STATEMENT

In our work, we aim at Neyman-orthogonal CATE estimation under differential privacy. Specifically, we aim to derive an $(\varepsilon, \delta)$-differentially private version of $\hat{g}_D(\cdot; \eta) = \arg\min_{g \in \mathcal{G}} R_D(g, \eta, \lambda(\pi))$ of the form

$$\hat{g}_{\mathrm{DP}}(\mathbf{x}; \eta) = \hat{g}_D(\mathbf{x}; \eta) + r(\varepsilon, \delta, \hat{g}_D, \eta) \cdot \mathbf{U}, \tag{8}$$

where $\hat{g}_D(\mathbf{x}; \eta) = (\hat{g}_D(x_1; \eta), \ldots, \hat{g}_D(x_d; \eta))^\top$, $\mathbf{U} \sim \mathcal{N}(0, \boldsymbol{I}_d)$, and where $r(\cdot)$ is a *calibration function*. Importantly, we consider *arbitrary* working model classes $\mathcal{G}$.

Our problem statement – and thus our framework – is intentionally flexible. (1) We assume that the propensity score is *not* known and, instead, is estimated from observational data. (2) We focus on Neyman-orthogonal meta-learners because these are *model-agnostic* and can thus be seamlessly instantiated with any machine learning model, including neural networks. (3) Our derivations are general and, therefore, apply to *any* orthogonal loss of the form in Eq. (3). Below, we present our DP-CATE framework for the R-learner due to its state-of-the-art performance. We additionally provide an extension of our framework for the DR-learner in Supplement B.

The above task is highly *non-trivial* as the identification of CATE relies on the standard causal assumptions of positivity, consistency, and unconfoundedness (Rubin, 2005). Yet, privacy mechanisms perturb different parts of the data (or the model), which could arguably violate the causal assumptions (in the case of the input perturbation) or lead to *biased* estimates (in the case of the model perturbation). Thus, instead of blindly introducing noise to the estimation setup to guarantee DP, we must cautiously calibrate the noise in a targeted setup to retain the consistency and quasi-oracle efficiency of the privatized CATE estimators.

## 4 OUR FRAMEWORK: DP-CATE

**Overview:** We now present our DP-CATE framework aimed at CATE estimation for Neyman-orthogonal meta-learners under $(\varepsilon, \delta)$-*differential privacy*. To address the challenges from above, we employ output perturbation, which is highly suitable to our purpose for two reasons. First, it allows us to ensure that causal assumptions are fulfilled even after perturbation. Second, we retain the favorable Neyman-orthogonality properties of the existing CATE estimation methods.

**Use cases:** Our framework comes in two variants, relevant for different use cases in medical practice:

(1) DP-CATE **for finite queries** ($\rightarrow$ **Sec. 4.1**): We aim to report a number $d$ of CATE estimates (e.g., treatment effects across different age groups). $\Rightarrow$ *How do we solve this?* We draw upon the shared property of robustness and privacy of ML models in terms of insensitivity to outliers and small measurement errors (Dwork & Lei, 2009). We then propose calibrating the noise added with a function $r(\cdot)$ that depends on the meta-learner's second-stage influence function. Importantly, our model-agnostic approach preserves the quasi-oracle guarantees of the non-private model stemming from the Neyman-orthogonal loss function.

(2) DP-CATE **for functional queries** ($\rightarrow$ **Sec. 4.2**): We release an estimate $\hat{g}_{\mathrm{DP}}$ of the complete CATE function $\tau$, which can then be queried arbitrarily often (e.g, as in clinical decision support systems). Existing output perturbation mechanisms only apply to scalar or finite-dimensional vector-valued outputs. Therefore, the above is only valid if the overall number of queries made to the algorithm is both finite and known before the perturbation. $\Rightarrow$ *How do we solve this?* We derive an output perturbation method based on Gaussian processes that is valid for all functional CATE estimates solving Eq. (3), as long as the estimation in the second stage is performed through a Gaussian kernel regression.

### 4.1 DP-CATE FOR A FIXED NUMBER OF QUERIES

Here, a total number of $d$ CATE estimates should be released. The number of queries, $d$, to the CATE function is known a priori. For notational simplicity, we thus rewrite the $d$ separate CATE estimates as a $d$-dimensional vector. We employ bold letters in the following to emphasize that we are interested in a vectorized version of the CATE meta-learner output $\hat{g}_D(\cdot; \eta)$, i.e., $\hat{g}_D(\mathbf{x}; \eta) \in \mathbb{R}^d$.

We now derive a calibration function $r(\cdot)$ that is applicable to any Neyman-orthogonal CATE meta-learner. We employ $r(\cdot)$ to calibrate a noise vector $\mathbf{U}$ with respect to the privacy budget $(\varepsilon, \delta)$ and the model sensitivity. Finally, we perturb $\hat{g}_D(\mathbf{x}; \eta)$ to fulfill DP through $\hat{g}_{\mathrm{DP}}(\mathbf{x}; \eta) = \hat{g}_D(\mathbf{x}; \eta) + r(\varepsilon, \delta, \hat{g}_D, \eta) \cdot \mathbf{U}$.

For this, we borrow ideas from the literature that observed similarities between differential privacy and robust statistics (e.g., Avella-Medina, 2021; Dwork & Lei, 2009) but which we carefully tailor to our setting in the following. Our idea is to employ the so-called *influence function* (IF) of the second-stage CATE model to calibrate the noise. The IF allows us to quantify how a single observation in $D$ influences CATE estimation and the model output. Intuitively, the IF describes the effect of an infinitesimally small perturbation of the input $z$ on the model output.

**Definition 3.** *Let $T$ be a functional of a distribution that defines the parameter of interest, $T = T(P) \in \mathbb{R}^d$. The* gross-error sensitivity *of $T$ at $z$ under $P$ is given by the supremum of the $l_2$ norm of the influence function, i.e.,*

$$\gamma(T, P) := \sup_{z \in \mathcal{Z}} \|\mathrm{IF}(z, T; P)\|_2, \tag{9}$$

*where $\mathrm{IF}(z, T; P) = \frac{\mathrm{d}}{\mathrm{d}t}\big[T((1-t)P + t\delta_z)\big]\big|_{t=0}$ is an influence function (IF) of $T$ at $z$ under a distribution with the density/probability mass function $P$, and $\delta_z$ denotes the Dirac-delta function.*

Next, we derive the IF of the second-stage models for CATE meta-learners. Observe that $T(P)$ depends on the data distribution directly through $P$ and indirectly through the first-stage estimation. Specifically, now $T(P) = g^*(\mathbf{x}; \eta)$ and $T(D) = \hat{g}_D(\mathbf{x}; \eta)$ if the nuisance functions were known; and $T(P) = g^*(\mathbf{x}; \hat{\eta}_{\tilde{D}})$ and $T(D) = \hat{g}_D(\mathbf{x}; \hat{\eta}_{\tilde{D}})$ if the nuisance functions are estimated.

We now state our main theorem for differentially private CATE estimation with a known number $d$ of queries. The intuition behind constructing the calibration function $r(\cdot)$ builds upon a result in Nissim et al. (2007) in which $(\varepsilon, \delta)$-differential privacy is achieved through calibrating noise with respect to the smooth sensitivity of the prediction model in comparison to the commonly employed global sensitivity from Definition 2. However, calculating the smooth sensitivity is still difficult or even infeasible for general function classes. Nevertheless, we show that the smooth sensitivity of the second-stage model can be upper bounded by the gross-error sensitivity $\gamma$ of the second-stage regression.[4]

**Theorem 1** (DP-CATE for finite queries)**.** *Let $z := (a, x, y)$ define a data sample following the joint distribution $\mathcal{Z}$ and $\hat{\eta}_{\mathrm{DP}} = (\hat{\mu}_{\mathrm{DP}}, \hat{\pi}_{\mathrm{DP}})$ the nuisance functions estimated on dataset $\tilde{D}$ in a $(\varepsilon/2, \delta/2)$-differentially private manner of choice.[5] Furthermore, let $D$ be the training dataset for the second-stage model with $|D| = n$. For $z = (a, x, y) \in \mathcal{Z}$, we define*

$$\hat{g}_{\mathrm{DP}}(\mathbf{x}; \hat{\eta}_{\mathrm{DP}}) := \hat{g}_D(\mathbf{x}; \hat{\eta}_{\mathrm{DP}}) + \underbrace{\gamma(T, D) \cdot c(\varepsilon, \delta, n)}_{r(\varepsilon, \delta, \hat{g}_D, \hat{\eta}_{\mathrm{DP}})} \cdot \mathbf{U}, \qquad T(P) = g^*(\mathbf{x}; \hat{\eta}_{\mathrm{DP}}), \tag{10}$$

$$\gamma(T, P) = \sup_{z \in \mathcal{Z}} \left\| h(g^*, \mathbf{x}, x, \hat{\eta}_{\mathrm{DP}})\, \rho(a, \hat{\pi}_{\mathrm{DP}}(x))\big(\phi(z, \hat{\eta}_{\mathrm{DP}}, \lambda(\hat{\pi}_{\mathrm{DP}}(x))) - g^*(x; \hat{\eta}_{\mathrm{DP}})\big) \right\|_2, \tag{11}$$

*where $\gamma(T, D)$ is a sample gross-error sensitivity with $T(D)$ instead of $T(P)$ in Eq. (9), $\mathbf{U} \sim \mathcal{N}(0, \mathbf{I}_d)$, $c(\varepsilon, \delta, n) := 5\sqrt{2\ln(n)\ln(2/\delta)}/(\varepsilon n)$, and where $h(g^*, \mathbf{x}, z, \hat{\eta}_{\mathrm{DP}}) \in \mathbb{R}^d$ and $g^*(\cdot; \hat{\eta}_{\mathrm{DP}})$ depend on the machine learning model employed for the second-stage regression. Then, $\hat{g}_{\mathrm{DP}}(\mathbf{x}; \hat{\eta}_{\mathrm{DP}})$ is $(\varepsilon, \delta)$-differentially private.*

*Proof.* We prove Theorem 1 in Supplement F. To do so, we first state the IF of the second-stage model that minimizes a weighted loss. Then, we calculate the gross-error sensitivity to show that the sample-size-weighted sensitivity upper-bounds the smooth sensitivity of the respective learner. $\square$

---

[4]See Lemma 5 in Supplement E

[5]As we need to query the nuisance functions multiple times, the privatization method needs to be suitable for functions, such as gradient perturbation through DP-SGD (Abadi et al., 2016).

In the case of the R-learner, Theorem 1 yields the following gross-error sensitivity $\gamma^{\mathrm{R}}(T, P)$:

$$\gamma^{\mathrm{R}}(T, P) = \sup_{z \in \mathcal{Z}} \left\| h(g^*, \mathbf{x}, x, \hat{\eta}_{\mathrm{DP}}) \underbrace{(a - \hat{\pi}_{\mathrm{DP}}(x))^2}_{\rho(a, \hat{\pi}_{\mathrm{DP}})} \left( \underbrace{\frac{y - \hat{\mu}_{\mathrm{DP}}(x, a)}{a - \hat{\pi}_{\mathrm{DP}}(x)} + \hat{\mu}_{\mathrm{DP}}(x, 1) - \hat{\mu}_{\mathrm{DP}}(x, 0)}_{\phi(z, \hat{\eta}_{\mathrm{DP}}, \lambda(\hat{\pi}_{\mathrm{DP}}))} - g^*(x; \hat{\eta}_{\mathrm{DP}}) \right) \right\|_2 . \quad (12)$$

In the following, we present two corollaries stating the form of the function $h(g^*, \mathbf{x}, z, \hat{\eta}_{\mathrm{DP}})$ for parametric models such as neural networks, as well as the non-parametric kernel ridge regression estimator.

**Corollary 1** (Parametric second-stage regression). *If the second-stage regression is a smooth parametric model, namely $\mathcal{G} = \{g(x; \theta) : \mathcal{X} \mapsto \mathbb{R}, \theta \in \Theta \subseteq \mathbb{R}^p\}$, then, in Theorem 1, we have*

$$g^*(\cdot; \hat{\eta}_{\mathrm{DP}}) = g(\cdot; \theta^*) \quad and \quad h(g^*, \mathbf{x}, x, \hat{\eta}_{\mathrm{DP}}) = 2 J_\theta \left[ g^*(\mathbf{x}; \hat{\eta}_{\mathrm{DP}}) \right] \cdot H_\theta^{-1} \cdot \nabla_\theta \left[ g^*(x; \hat{\eta}_{\mathrm{DP}}) \right], \quad (13)$$

*where $\theta^* = \arg \min_{\theta \in \Theta} R_P(g, \hat{\eta}_{\mathrm{DP}}, \lambda(\hat{\pi}_{\mathrm{DP}}))$; $J_\theta \in \mathbb{R}^{d \times p}$ is a Jacobian matrix wrt. $\theta$; $H_\theta = \nabla_\theta^2 \left[ R_P(g^*, \hat{\eta}_{\mathrm{DP}}, \lambda(\hat{\pi}_{\mathrm{DP}})) \right] \in \mathbb{R}^{p \times p}$ is a Hessian matrix; and $\nabla_\theta$ is a gradient.*

*Proof.* We prove Corollary 1 in Supplement F. $\square$

**A note on applicability:** Corollary 1 requires access to and invertibility of $H_\theta$. This might restrict the applicability of certain deep neural networks for the second-stage regression. However, we note that, in Theorem 1, we are only interested in predicting the CATE at certain values $\mathbf{x}$. This motivates the local CATE estimation through kernel weighting. We present our alternative approach below.

**Corollary 2** (Non-parametric second-stage regression). *If the second-stage regression is a kernel ridge regression with $\Lambda(g) = \lambda \|g\|_{\mathcal{H}}^2$, where $\mathcal{G} = \mathcal{H}$ is a reproducing kernel Hilbert space (RKHS) induced by a kernel $K(\cdot, \cdot) : \mathcal{X} \times \mathcal{X} \mapsto \mathbb{R}^+$, then, in Theorem 1, we have*

$$g^*(\cdot; \hat{\eta}_{\mathrm{DP}}) = (\mathbb{L}_\rho + \lambda \mathbb{I})^{-1} S(\cdot) \quad and \quad h(g^*, \mathbf{x}, x, \hat{\eta}_{\mathrm{DP}}) = (\mathbb{L}_\rho + \lambda \mathbb{I})^{-1} K(\cdot, x)(\mathbf{x}), \quad (14)$$

*where $\mathbb{L}_\rho : \mathcal{H} \mapsto \mathcal{H}$ is a weighted covariance operator $\mathbb{L}_\rho f(\cdot) = \mathbb{E}\left[ \rho(A, \hat{\pi}_{\mathrm{DP}}(X)) K(\cdot, X) f(X) \right]$; $\lambda \mathbb{I} f(\cdot) = \lambda f(\cdot)$ is a scaling operator; $S \in \mathcal{H}$ is a cross-covariance functional $S(\cdot) = \mathbb{E}\left[ \rho(A, \hat{\pi}_{\mathrm{DP}}(X)) K(\cdot, X) \phi(Z, \hat{\eta}_{\mathrm{DP}}, \lambda(\hat{\pi}_{\mathrm{DP}}(X))) \right]$.*

*Proof.* We prove Corollary 2 in Supplement F. $\square$

**Scalability:** The complexity of calculating $\gamma(\cdot)$ is *independent* of the size of the dataset once the second-stage model $\hat{g}_D(\cdot; \hat{\eta}_{\mathrm{DP}})$ is fitted.

**Theorem 2** (Neyman-orthogonality and quasi-oracle efficiency of DP-CATE). *The privatization of the second-stage model asymptotically preserves the property of Neyman-orthogonality, namely*

$$\|g^*(\cdot; \eta) - \hat{g}_{\mathrm{DP}}(\cdot; \hat{\eta}_{\mathrm{DP}})\|_{L_2}^2 \lesssim R_P\left(g^*(\cdot; \hat{\eta}_{\mathrm{DP}}), \hat{\eta}_{\mathrm{DP}}, \lambda(\hat{\pi}_{\mathrm{DP}})\right) - R_P\left(g^*(\cdot; \eta), \hat{\eta}_{\mathrm{DP}}, \lambda(\hat{\pi}_{\mathrm{DP}})\right) + R_2(\hat{\eta}_{\mathrm{DP}}, \eta)$$
$$+ \underbrace{\|g^*(\cdot; \hat{\eta}_{\mathrm{DP}}) - \hat{g}_D(\cdot; \hat{\eta}_{\mathrm{DP}})\|_{L_2}^2}_{\text{depends on the model class } \mathcal{G}} + \underbrace{o_P(n^{-1})}_{\text{output perturbation}}. \quad (15)$$

*Furthermore, under additional regularity conditions on the privatization of the nuisance functions (e.g., gradient perturbation), our DP-CATE achieves quasi-oracle efficiency. Specifically, if the original estimation of the nuisance functions is at rate of at least $o_P(n^{-1/4})$, then the privatized estimation preserves this rate.*

*Proof.* We prove Theorem 2 in Supplement F. $\square$

## 4.2 DP-CATE FOR COMPLETE CATE FUNCTIONS

In this variant of DP-CATE, we seek to privately release an estimate $\hat{g}_D(\cdot; \eta)$ of the complete CATE function $\tau(\cdot)$. Note that we cannot leverage Theorem 1 because it is only applicable to CATE estimates but not to complete functions. Instead, we must now derive a tailored approach.

Intuitively, we need to find (I) a type of noise and (II) a calibration function that does not depend on $d$. More precisely, the added noise should be a function itself to guarantee the privacy of the CATE function. For (I), we propose to add a calibrated Gaussian process (GP) to the predicted CATE. [6]

---

[6] We refer to Rasmussen & Williams (2006) for an in-depth introduction to Gaussian processes.

**Definition 4** (Gaussian process). *A family of random variables $\{X_t\}_{t \in T}$ is a Gaussian process if for any subset $S \in T$, $\{X_t\}_{t \in S}$ has a Gaussian distribution. The process is entirely determined by its mean function $m(t) := \mathbb{E}[X_t]$ and covariance function $K(s, t) := \mathrm{Cov}(X_s, X_t)$.*

We are left with answering question (II) from above: *how to calibrate the GP noise to fulfill DP?* We make the following important observation: If $\hat{g}_D(\cdot; \eta)$ lies in a reproducing kernel Hilbert space (RKHS), we can calibrate the GP noise with respect to the RKHS norm using results from functional analysis (Hall et al., 2013). To ensure that $\hat{g}_D(\cdot; \eta)$ indeed lies in an RKHS, we can later follow prior research (e.g., Kennedy, 2023a; Singh et al., 2024) and model the second-stage estimation in DP-CATE framework as a Gaussian kernel regression.

We now want to bound the difference of CATE functions trained on neighboring datasets with respect to the norm of the Hilbert space. Recall that differential privacy of function $f$ under the Gaussian mechanism in Def. 2 requires knowledge about $\sup_{D \sim D', \mathbf{x} \in \mathcal{X}^d} ||\hat{f}_D(\mathbf{x}) - \hat{f}_{D'}(\mathbf{x})||_2$ to calibrate the Gaussian noise variable. Similarly, we now require knowledge of $\sup_{D \sim D'} ||\hat{f}_D - \hat{f}_{D'}||_{\mathcal{H}}$ where $\hat{f}_D$ specifies an RKHS regression to calibrate the Gaussian process. However, to the best of our knowledge, no closed-form solution for this quantity exists. We thus derive the following lemma as an extension of Hall et al. (2013) in our setting.

**Lemma 1.** *Let $\mathcal{H}$ denote the RKHS induced by the Gaussian kernel $K(x, x') = (\sqrt{2\pi}h)^{-q} \exp(-\|x - x'\|_2^2 / (2h^2))$ for $x, x' \in \mathcal{X} \subseteq \mathbb{R}^q$, and let $\hat{f}_D$ be the optimal solution to the RKHS regression*

$$\hat{f}_D(\cdot) = \arg\min_{f \in \mathcal{H}} \frac{1}{n} \sum_{i=1}^n w(X_i) \cdot \ell(f(X_i), Y_i) + \lambda \|f\|_{\mathcal{H}}^2, \tag{16}$$

*where $w(\cdot) > 0$ is a weight function, $D$ is a dataset with $|D| = n$, and $\ell(\hat{y}, y)$ is a convex and Lipschitz loss function in $\hat{y}$ with Lipschitz constant $L$. Then, for $D \sim D'$, we have*

$$||\hat{f}_D - \hat{f}_{D'}||_{\mathcal{H}} \leq \sup_{x \in \mathcal{X}} [w(x)] \frac{L}{\lambda n} \left( \sqrt{(2\pi)}h \right)^{-q}. \tag{17}$$

*Proof.* We prove Lemma 1 in Supplement F. $\square$

We now use the above results and present our DP framework for CATE functions: (i) Stage 1: We estimate the $(\varepsilon/2, \delta/2)$-differentially private nuisance functions $\hat{\mu}_{\mathrm{DP}}$ and $\hat{\pi}_{\mathrm{DP}}$ through *any* parametric or non-parametric machine learning method and perturbation method. (ii) Stage 2: We perform a Gaussian kernel regression to minimize Eq. (3). (iii) We calibrate a suitably chosen Gaussian process based on Lemma 1 and add the resulting GP to the CATE function. In the case $\ell$ is a squared loss, the quasi-oracle efficiency of our framework directly follows from Theorem 2 and (Foster & Syrgkanis, 2019). We present the pseudo-code for DP-CATE in Supplement C.

**Theorem 3** (DP-CATE for functional queries). *Let $\hat{\mu}_{\mathrm{DP}}$ and $\hat{\pi}_{\mathrm{DP}}$ denote the $(\varepsilon/2, \delta/2)$-differentially private nuisance estimators trained in stage 1 on $\tilde{D}$. Let $z = (a, x, y)$ be a data sample from dataset $D$ with $|D| = n$ and $x \in \mathcal{X} \subseteq \mathbb{R}^q$. Let $\mathcal{H}$ denote the RKHS induced by the kernel $K(x, x') = (\sqrt{2\pi}h)^{-q} \exp(-\|x - x'\|_2^2 / 2h^2)$, and let $\ell(\cdot, \cdot)$ be a convex and Lipschitz loss function with Lipschitz constant $L$. We define $\hat{g}_D(\cdot; \hat{\eta}_{\mathrm{DP}})$ as the second-stage regression solving Eq. (3) via*

$$\hat{g}_D(\cdot; \hat{\eta}_{\mathrm{DP}}) = \arg\min_{g \in \mathcal{H}} \frac{1}{n} \sum_{i=1}^n \rho(A_i, \hat{\pi}_{\mathrm{DP}}(X_i)) \, \ell\big(g(X_i), \phi(Z_i, \hat{\eta}_{\mathrm{DP}}, \lambda(\hat{\pi}_{\mathrm{DP}}(X_i)))\big) + \lambda \|g\|_{\mathcal{H}}^2. \tag{18}$$

*Furthermore, let $U(\cdot) \in \mathcal{H}$ be the sample path of a zero-centered Gaussian process with covariance function $K(x, x')$. Then, $(\varepsilon, \delta)$-differential privacy is guaranteed by*

$$\hat{g}_{\mathrm{DP}}(\cdot; \hat{\eta}_{\mathrm{DP}}) := \hat{g}_D(\cdot; \hat{\eta}_{\mathrm{DP}}) + \underbrace{\sup_{(a,x) \in \{0,1\} \times \mathcal{X}} \big[\rho(a, \hat{\pi}_{\mathrm{DP}}(x))\big] \frac{4L\sqrt{2\ln(2/\delta)}}{(\sqrt{2\pi}h)^q \lambda n \varepsilon}}_{r(\varepsilon, \delta, \hat{g}_D, \hat{\eta}_{\mathrm{DP}})} \cdot U(\cdot). \tag{19}$$

*Proof.* We prove Theorem 3 in Supplement F. $\square$

Importantly, for both R- and DR-learner, $\sup_{(a,x)\in\{0,1\}\times\mathcal{X}}\left[\rho(a,\hat{\pi}_{DP}(x))\right]\leq 1$. Also, the above theorem requires a convex Lipschitz loss $\ell(\cdot,\cdot)$. There are many suitable loss functions (e.g., the squared loss on bounded domains, a trimmed squared loss, or the Huber loss). For many losses, the Lipschitz constant is data-independent and directly computable from the loss function.[7] We further require the second-stage in the meta-learner to be a Gaussian kernel regression, which is widely used in causal inference (e.g., Kennedy, 2023a; Singh et al., 2024). Nevertheless, our DP-CATE is still fairly flexible in that any Neyman-orthogonal meta-learner can be used (e.g., R-learner, DR-learner) and that *any* machine learning model can be used for nuisance estimation. In Supplement C, we present an algorithm for releasing private outputs of the function $\hat{g}_{DP}(\cdot;\hat{\eta}_{DP})$.

## 5 EXPERIMENTS

**Implementation:** Our DP-CATE is model-agnostic and highly flexible. Therefore, we instantiate our DP-CATE with multiple versions of the R-leaner (Nie et al., 2021) where we vary the underlying base learners. Hence, we implement the pseudo-outcome regression in the second stage via both a neural network (NN) and the Kernel regression estimator (KR). We estimate the nuisance functions through neural networks. This is recommended as one typically allows for flexibility in the nuisance functions (Curth & van der Schaar, 2021). Details on implementation and training are in Supplement G. We emphasize again that there are no suitable baselines for our task.

**Performance metrics:** As explained in Sec. 2, there are no flexible CATE meta-learners ensuring DP. Thus, there is no suitable baseline for our task. Hence, we perform experiments to primarily show the applicability of our DP-CATE across different privacy budgets. We expect that the prediction performance will increase with increasing privacy budget and then approach the prediction performance of a non-private CATE learner. We measure the performance via the *precision in estimation of heterogeneous effects* (PEHE) with regard to the true CATE (e.g., Hill, 2011).

### 5.1 EVALUATION ON SYNTHETIC DATASETS

**Synthetic datasets:** Due to the fundamental problem of causal inference, counterfactual outcomes are never observed in real-world data. Thus, we follow common practice and evaluate DP-CATE on synthetic data, which allows us to access the ground-truth CATE and compute the PEHE (e.g., Kennedy, 2023a; Oprescu et al., 2019). We evaluate DP-CATE on 300 queries.

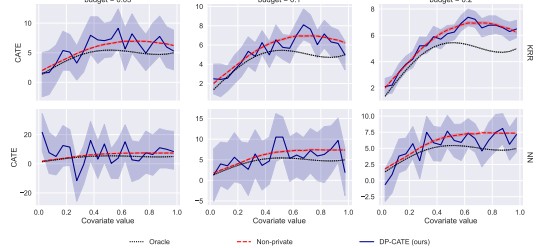

Figure 3: **Dataset 1** (finite queries). Predictions under different base learners and privacy budgets.

We consider two settings with different treatment effect complexities following Oprescu et al. (2019). ● **Dataset 1** contains two observed confounders from which only one influences the CATE. This allows us to visualize the CATE function and the effect of privatization on the prediction for varying covariate values. ● **Dataset 2** contains 30 observed confounders and multiple dimensions influence the CATE. Details are in Supplement G.

**Results for finite queries:** ● **Dataset 1**: Fig. 3 shows the predictions for different base learners and different privacy budgets. We make the following observations: (1) Our DP-CATE performs as expected: with increasing privacy budget, the predictions become less 'noisy' and converge to those of the non-private estimator. (2) Our DP-CATE shows consistent patterns for

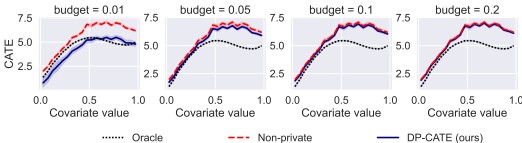

Figure 4: **Dataset 1** (functional queries). Predictions under different privacy budgets.

different base learners. For example, the predictions under both KR and NN are almost identical, showing the flexibility and robustness of our framework. ● **Dataset 2**: Fig. 5 shows the PEHE. Note that we directly compare DP-CATE for finite queries (combined with KR and NN) and DP-CATE for functional queries. Again, we find that our DP-CATE performs as expected: the PEHE decreases with increasing privacy budget and converges towards that of the non-private learner.

---

[7] For example, for the $l_1$ loss, the Lipschitz constant $L$ equals 1; for the Huber loss, $L$ equals the loss parameter $\delta$; and, for the truncated $l_2$ loss, the constant equals the gradient at the truncation value.

**Results for functional queries:** • **Dataset 1**: Fig. 4 shows the predictions of DP-CATE for functional queries across different privacy budgets. We observe similar behavior as in the case of finite queries: With increasing privacy budget, the predictions converge to those of the non-private learner. • **Dataset 2**: Fig. 5 shows the results for more complex dataset. As before, we observe that DP-CATE for functions behaves in the same way as DP-CATE for finite queries. Overall, our findings are robust across datasets and base learner specifications.

## 5.2 EVALUATION ON MEDICAL DATASETS

**Medical datasets:** We demonstrate the applicability of DP-CATE to medical datasets by using the **MIMIC-III** dataset (Johnson et al., 2016) and the **TCGA** dataset (Weinstein et al., 2013). MIMIC-III contains real-world health records from patients admitted to intensive care units at large hospitals. We aim to predict a patient's red blood cell count after being treated with mechanical ventilation. The Cancer Genome Atlas (TCGA) dataset contains a large collection of gene expression data from patients with different cancer types. We assign a treatment indicator based on the gene expression level and aim to predict a constant effect across all expression levels. Details are in Supplement G.

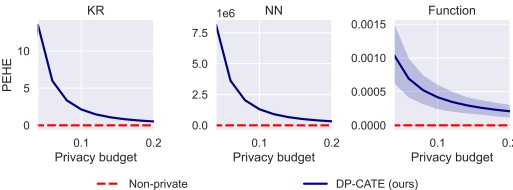

Figure 5: **Dataset 2**. Prediction errors under different privacy budgets of DP-CATE (finite) on KR and NN and DP-CATE (functional) over 10 runs. Plots centered at the PEHE of the original learner.

**Results:** • **MIMIC-III:** Fig. 6 reports the predictions of the CATE against different levels of hematocrit and different privacy budgets. Here, we have $d = 1312$ queries (i.e., the size of the test set). Our DP-CATE framework works as desired: for smaller privacy budgets, more noise should be added, which is also reflected in a larger variation of the predictions. • **TCGA:** Fig. 7 shows again that our DP-CATE is effective on a large number of queries (i.e., $d = 2659$). Overall, the loss in precision due to DP (i.e., when comparing DP-CATE to the non-private learner) is fairly small.

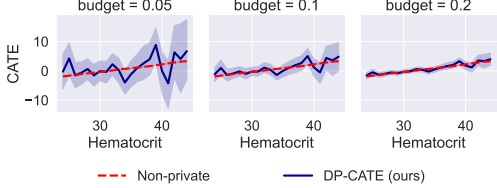

Figure 6: **MIMIC-III** (finite queries). Our DP-CATE generates private estimates of the effect of ventilation for different levels of hematocrit.

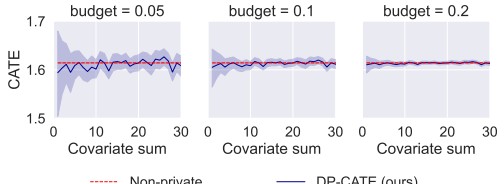

Figure 7: **TCGA** (finite queries). DP-CATE consistently estimates the constant treatment effect across the sum of all covariate values.

**Takeaways:** *The prediction error of DP-CATE decreases with larger privacy budgets as desired and converges to the non-private error, confirming that our framework makes precise CATE predictions.*

## 6 DISCUSSION

**Applicability:** We provide a general framework for differentially private and Neyman-orthogonal CATE estimation from observational data. First, our DP-CATE is carefully designed for observational data, which are common in medical applications (Feuerriegel et al., 2024). Second, DP-CATE is applicable to various meta-learners (e.g., R-learner, DR-learner), which are widely used in practice. Third, DP-CATE allows different use cases: one can release either a certain number of CATE estimates or even the complete CATE function (e.g., as in clinical decision support systems).

**Extension to the DR-learner:** Our derivations focus on the popular R-learner due to its favorable theoretical properties. Nevertheless, our DP-CATE can be applied to any other Neyman-orthogonal meta-learner. In Supplement B, we thus provide an extension to the DR-learner. Therein, we also provide additional numerical experiments. At a technical level, the weight function for the DR-learner simplifies to $\lambda^{\mathrm{DR}}(\pi(x)) = 1$ in contrast to the weight function of the R-learner $\lambda^{\mathrm{R}}(\pi(x)) = \pi(x)(1 - \pi(x))$. However, $\lambda^{\mathrm{DR}}(\pi(x))$ is less suitable for DP because it needs a larger noise term during the perturbation (see Supplement B for a detailed discussion), which hinders downstream performance. Hence, we recommend employing DP-CATE with the R-learner in practice.

**Conclusion:** Ensuring the privacy of sensitive information in treatment effect estimation is mandated for ethical and legal reasons. Here, we provide the first framework for differentially private CATE estimation from observational data using meta-learners.

ACKNOWLEDGEMENTS

We thank Dennis Frauen and Lars van der Laan for their helpful feedback on our manuscript. Our research was supported by the DAAD program Konrad Zuse Schools of Excellence in Artificial Intelligence, sponsored by the Federal Ministry of Education and Research.

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

## A    ADDITIONAL BACKGROUND MATERIAL

### A.1    DIFFERENTIAL PRIVACY

**DP mechanisms:**  There are four main strategies of DP mechanisms (see Fig. 8): (i) *Input perturbation* independently randomizes each data sample before model training (e.g., Fukuchi et al., 2017). (ii) *Objective perturbation* adds a random term to the objective and releases the respective minimizer (e.g., Iyengar et al., 2019; Kifer et al., 2012; Redberg et al., 2023). The mechanisms in this

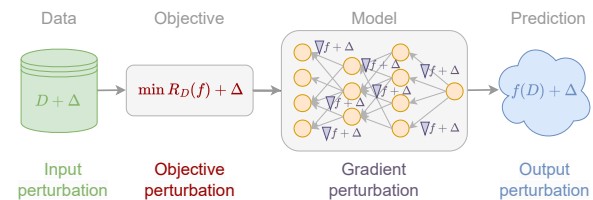

Figure 8:  Privacy mechanisms in the machine learning workflow.

field commonly make strong assumptions on the smoothness or convexity of the objective. (iii) *Gradient perturbation* clips, aggregates, and adds noise to the gradient updates in each step of gradient descent methods during model training (e.g., Abadi et al., 2016; Wang et al., 2017; 2019). (iv) *Output perturbation* adds noise to the non-private model prediction before its release (e.g., Chaudhuri et al., 2011; Zhang et al., 2022). All stated mechanisms are general strategies that must be carefully adapted to our CATE estimation setting. In our work, we employ output perturbation for the reasons explained below.

**Choice of DP mechanism:** Our DP-CATE framework employs *output perturbation* to achieve DP. Output perturbation is highly suitable for our setting since (i) it ensures that causal assumptions are fulfilled even after perturbation, and (ii) it retains the power of existing CATE estimation methods for addressing the fundamental problem of causal inference. The other DP strategies discussed above fail to fulfill the requirements.

In contrast, input perturbation might introduce confounding bias or violate the consistency assumption. For gradient and objective perturbation, the convergence of the model might be unclear. Furthermore, objective perturbation might fail to achieve the targeted privacy guarantee if the model does not converge to the exact global minimum in finite time (Iyengar et al., 2019). Gradient perturbation results in a non-trivial privacy overhead and does not align with our goal of providing a model-agnostic meta-learning framework (Redberg et al., 2023).

### A.2    EXTENDED RELATED WORK

The only existing method designed for our setting was proposed by Niu et al. (2022). The authors provide an algorithm for differentially private CATE estimation through existing CATE meta-learners. However, the method necessitates special private base learners for the separate sub-algorithms in each stage of the meta-learner. It is thus *not* agnostic to the choice of ML method for the first- and second-stage regressions, meaning that arbitrary choices are *not* supported. Furthermore, it has been shown that privatizing different parts of causal estimators separately can result in biased causal estimates  (Ohnishi & Awan, 2023).

A different line of work proposes *locally differentially private* (LDP) algorithms (Agarwal & Singh, 2024; Huang & Ascara, 2023; Ohnishi & Awan, 2023). This notion of privacy becomes necessary if the central data curator cannot be trusted. During data collection, calibrated noise is added to each sample before adding it to the database. However, the perturbed data might violate the assumptions to identify causal treatment effects. Furthermore, this notion of privacy significantly reduces the predictive accuracy of the estimators (Huang & Ascara, 2023). Thus, whenever the data curator is a trusted party (as assumed in our work), global differential privacy is sufficient and should be the notion of choice as it is less accuracy-compromising than its local counterpart.

### A.3    FUTURE RESEARCH DIRECTIONS FOR PRIVACY IN CAUSAL ML

Applying causal machine learning methods to real-world problems requires the methods to adhere to guidelines on ML safety and ethical ML. The privacy of predictions is only one aspect of the former. Of note, privacy has a direct effect on other ethical and safety-related aspects, such as the

uncertainty or the fairness in the predictions. Hence, there are many impactful directions for future research.

Uncertainty in causal ML, including causality-specific types of uncertainty such as unobserved confounding, have been studied in the literature (e.g. Jesson et al., 2020; Melnychuk et al., 2024; Schröder et al., 2024a). However, so far, there is no method that quantifies the uncertainty added through the privatization of the causal model or the predictions. Developing such methods is an important step for future research.

Ethical causal ML has been mostly studied through the lens of causal fairness (e.g., Frauen et al., 2024; Ma et al., 2023; Plecko & Bareinboim, 2024; Schröder et al., 2024b). However, the trade-off between fair and private causal ML is unexplored. Research outside the field of causal ML suggests opposing effects of privatization and fairness (e.g., Tran et al., 2021). Therefore, investigating this relationship is an interesting avenue for future research and may help in providing ethical, causal treatment effect estimation.

## A.4 Theory on CATE estimation

The estimation of causal quantities, such as the conditional average treatment effect $\tau(x) = \mathbb{E}[Y(1) - Y(0) \mid X = x]$, involves counterfactual quantities $Y(a)$, since only one outcome per individual can be observed. Here, $Y(a)$ is the potential outcome that would hypothetically be observed if a decision $a$ is taken.

Due to the above, identification of causal effects from observational data necessitates the following three assumptions that are common in the literature (e.g., Curth & van der Schaar, 2021; Feuerriegel et al., 2024):

1. Consistency: The potential outcome $Y(a)$ equals the observed factual outcome $Y$ when the individual was assigned treatment $A = a$.

2. Positivity/overlap: The treatment assignment is not deterministic. Specifically, there exists a positive probability for each possible combination of features to be assigned to both the treated and the untreated group, i.e., there exists $\kappa > 0$ such that $\kappa < \pi(x) < 1 - \kappa$ for all $x \in \mathcal{X}$.

3. Unconfoundedness: Conditioned on the observed covariates, the treatment assignment is independent of the potential outcomes, i.e., $Y(0), Y(1) \perp\!\!\!\perp A \mid X$. Specifically, there are no unobserved variables (confounders) influencing both the treatment assignment and the outcome.

Importantly, the above assumptions are standard in the literature. Further, the assumptions are necessary for consistent causal effect estimation for *any* machine learning model. Then, CATE is identifiable as

$$\tau(x) := \mathbb{E}[Y(1) - Y(0) \mid X = x] = \mu(x, 1) - \mu(x, 0), \tag{20}$$

where $\mu(x, a) = \mathbb{E}[Y \mid X = x, A = a]$. To estimate $\tau$, one could thus train a machine learning model that estimates the aforementioned conditional expectation and then calculates the difference in conditional expectations for a given $X = x$. This is commonly referred to as *plug-in* method, yet which has several drawbacks, as we outline below. Rather, the preferred way to estimate the CATE is through meta-learners.

Meta-learners define model-agnostic algorithms, which can be implemented with arbitrary machine learning algorithms. Therefore, meta-learners are flexible and commonly employed in practice. CATE meta-learners can be classified into four different categories, depending on the ways they leverage the data: (1) one-step plug-in learners, (2) two-stage regression-adjusted learners, (3) two-stage propensity-weighted learners, and (4) two-stage Neyman-orthogonal learners (Curth & van der Schaar, 2021). Meta-learners for specialized tasks have also been proposed recently, such as partial identification or treatment effects over time (e.g., Frauen et al., 2025; Oprescu et al., 2023; Schweisthal et al., 2024)

We now discuss each type of meta-learner and their potential drawbacks in more detail:

1. *One-step plug-in learner:* Here, ML models are trained to predict $\mu(x, a)$, either one single model for both treatment values or two different models, i.e., $\hat{\mu}(x, 1)$ and $\hat{\mu}(x, 0)$. Then, the CATE is estimated directly as $\hat{\tau}(x) = \hat{\mu}(x, 1) - \hat{\mu}(x, 0)$.

2. *Two-stage regression-adjusted learner:* In the observed data, the difference between factual and counterfactual outcomes is never present. Therefore, two-stage learners construct *pseudo-outcomes* as surrogates, which equal the CATE in expectation. The regression-adjusted learner designs the pseudo-outcome through a reweighting based on the function $\mu$, which is estimated by $\hat{\mu}$ in the first stage. A misspecification of $\hat{\mu}$ results in a biased CATE estimator, where the error of $\hat{\mu}$ propagates with the same order into the final estimator $\hat{\tau}$.

3. *Two-stage propensity-weighted learner:* Here, the pseudo-outcome is constructed based on the Horvitz-Thompson transformation. Only the propensity function $\pi$ needs to be estimated in the first step by $\hat{\pi}$. A misspecification of $\hat{\pi}$ results in a biased CATE estimator. Here, again, the error of $\hat{\pi}$ propagates with the same order into the final estimator $\hat{\tau}$.

4. *Two-stage Neyman-orthogonal learners:* Different from the above two-stage learners, these learners have a *lower error* if either the propensity function $\pi$ or the outcome regressions $\mu$ are correctly specified. Specifically, the final estimation is first-order insensitive to small errors in the nuisance functions (known as quasi-oracle efficiency). This property is achieved through Neyman-orthogonal losses such as Eq. (3).

Hence, we focus on the two-stage Neyman-orthogonal learners throughout our work due to the clear advantages.

# B EXTENSION TO THE DR-LEARNER

In our main paper, we focused on the R-learner for the derivations and the experiments. However, DP-CATE is applicable to any weighted two-stage Neyman-orthogonal CATE meta-learner. Here, we now provide an extension to the DR-learner.

For the DR-learner, the weight function simplifies to $\lambda^{\mathrm{DR}}(\pi(x)) = 1$. Then, the $d$-dimensional $(\varepsilon, \delta)$-differentially private CATE estimated through the DR-learner is given by

$$\hat{g}_{\mathrm{DP}}(\mathbf{x}; \hat{\eta}_{\mathrm{DP}}) = \hat{g}_D(\mathbf{x}; \hat{\eta}_{\mathrm{DP}}) + \gamma^{\mathrm{DR}}(T, D) \cdot c(\varepsilon, \delta, n) \cdot \mathbf{U}, \tag{21}$$

where $\gamma^{\mathrm{DR}}(T, D)$ is a sample version of the gross-error sensitivity of $\gamma^{\mathrm{DR}}(T, P)$, $c(\varepsilon, \delta, n) := 5\sqrt{2\ln(n)\ln(2/\delta)}/(\varepsilon n)$, and $\mathbf{U} \sim \mathcal{N}(0, \boldsymbol{I}_d)$. Specifically, $\gamma^{\mathrm{DR}}(T, P)$ is given by

$$\gamma^{\mathrm{DR}}(T, P) = \sup_{z \in \mathcal{Z}} \left\| h(g^*, \mathbf{x}, x, \hat{\eta}_{\mathrm{DP}}) \underbrace{\left( \frac{(a - \hat{\pi}_{\mathrm{DP}}(x))(y - \hat{\mu}_{\mathrm{DP}}(x, a))}{\hat{\pi}_{\mathrm{DP}}(x)(1 - \hat{\pi}_{\mathrm{DP}}(x))} + \hat{\mu}_{\mathrm{DP}}(x, 1) - \hat{\mu}_{\mathrm{DP}}(x, 0) - g^*(x; \hat{\eta}_{\mathrm{DP}}) \right)}_{\phi(z, \hat{\eta}_{\mathrm{DP}}, \lambda(\hat{\pi}_{\mathrm{DP}}))} \right\|_2. \tag{22}$$

**Observation:** The noise calibration necessary for the privatization requires maximizing the influence function. For the DR-learner, the IF includes the inverse of $\pi(x)(1 - \pi(x))$. Although we assume $\pi(x)$ to be bounded away from zero and one (due to the overlap assumption), maximizing over this term can lead to a very large calibration factor. This might limit the applicability of the DR-learner for differentially private CATE estimation.

Below, we evaluate the private DR-learner in the same manner as the R-learner in the main paper. We observe the expected behavior in Fig. 9.

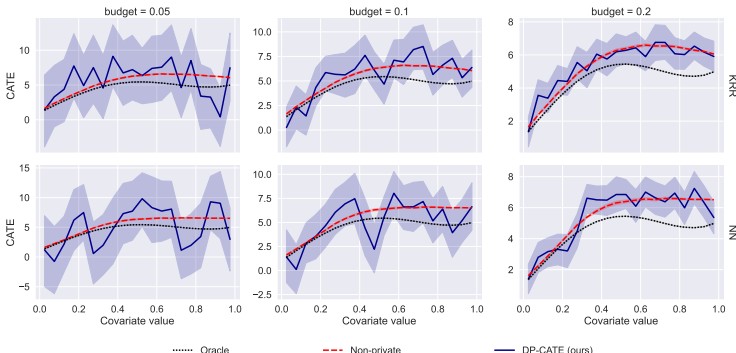

Figure 9: Evaluation of DP-CATE for finite queries the on DR-learner for different base-learner specifications on dataset 1.

## C   SIMILARITIES AND DIFFERENCES OF THE FINITE AND FUNCTIONAL DP-CATE FRAMEWORK

### C.1   DISCUSSION

If one only wants to release private CATE estimates once, both approaches ①  and ② are applicable. Nevertheless, the second approach, which we call the "functional approach", can also be employed for iteratively querying the function, which is especially of interest to medical practitioners aiming to assess the treatment effect of a drug for various patients with different characteristics. Put simply, the functional approach is relevant when companies want to release a decision support system to guide treatment decisions of individual patients. Since such treatment decisions are made based on the entire CATE model, the functional approach is preferred. In contrast, the first approach (which we call "finite-query approach") is preferred whenever only a few CATE values should be released. This is relevant for researchers (or practitioners) who may want to share the treatment effectiveness for a certain number of subgroups (but not for individual patients).

The functional approach requires sampling from a Gaussian process. Depending on whether one aims to report finitely many queries once through this approach or iteratively query the function, the sampling procedure from the Gaussian process $U(\cdot)$ differs. We highlight the differences in the type of queries in the following:

- **Simultaneous finitely many queries:** When querying the function only once with a finite number of queries, sampling from a Gaussian process implies sampling from the prior distribution of the Gaussian process. In empirical applications, this means that one samples from a multivariate normal distribution. Therefore, the noise added in the functional approach is similar to the finite-query approach. However, the approaches ① and ② are not the same, as the noise added in the functional approach is correlated, whereas the noise variables in the finite-query approach are independent. Therefore, the functional approach might result in a consistent under- or overestimation of the target. Still, both approaches guarantee privacy.

- **Iteratively querying the function:** In this setting, sampling from a Gaussian process implies sampling from the posterior distribution of the process. Specifically, if no query has been made to the private function yet, the finite-query approach proceeds by providing the first private CATE estimate of query $x_1$. Observe that the privatization of every further iterative query $x_i$ needs to account for the information leakage through answering former queries. Thus, sampling from a Gaussian process now relates to sampling from the posterior distribution. To do so, it is necessary to keep track of and store former queries $x_1, \ldots, x_{i-1}$ and the privatized outputs. This setting is entirely different from our finite setting approach, in which we propose adding Gaussian noise scaled by gross-error sensitivity.

### C.2   ALGORITHMS FOR DP-CATE FUNCTIONS

Private outputs of the function $\hat{g}_{\mathrm{DP}}(\cdot; \hat{\eta}_{\mathrm{DP}})$ in Theorem 3 can be released in two ways: (i) the standard *batch* setting presented in Algorithm 1, in which the private function outputs a private vector of CATE estimates *once* and (ii) the *iterative* (or *online*) setting, in which the function is queried iteratively, outputting one private CATE estimate at a time. Below, we provide an alternative algorithm to apply Theorem 3 in an iterative way.

---

**Algorithm 1:** Pseudo-code of out DP-CATE for functions

---

**Input:** CATE meta-learner $\hat{g}_D(\cdot; \hat{\eta}_{DP})$ trained on dataset $D$ with $|D| = n$, Gaussian kernel matrix $(K(x_i, x_j))_{i,j=1}^d$, query $\mathbf{x}_{query} \in \mathcal{X}^d$, privacy budget $\varepsilon$, $\delta$, Lipschitz const. $L$, ridge regularization $\lambda$

**Output:** Privatized CATE function $\hat{g}_{DP}(\cdot; \hat{\eta}_{DP})$

```
/* Calculate calibration term r                                      */
```
$r \leftarrow \sup_{(a,x) \in \{0,1\} \times \mathcal{X}} \left[ \rho(a, \hat{\pi}_{DP}(x)) \right] (4L\sqrt{2\log(2/\delta)})/((\sqrt{2\pi}h)^q \cdot \lambda n \varepsilon);$
```
/* Sample from Gaussian process                                      */
```
$\mathbf{U} \sim \mathcal{N}(\mathbf{0}_d, (K(x_i, x_j))_{i,j=1,\ldots,d});$
```
/* Return private estimates                                          */
```
$\hat{g}_{DP}(\mathbf{x}_{query}; \hat{\eta}_{DP}) \leftarrow \hat{g}_D(\mathbf{x}_{query}; \hat{\eta}_{DP}) + r \cdot \mathbf{U};$

---

*Iterative approach:* If no query has been made to the private function yet, we can employ Algorithm 2 to provide a private CATE estimate $\hat{g}_{DP}(x_1; \hat{\eta}_{DP})$, where $x_1$ denotes the first query. Specifically, Algorithm 2 samples from a Gaussian Process *prior* by sampling a suitable multivariate Gaussian noise variable. Observe that the privatization of every further iterative query $x_i$ needs to account for the information leakage by answering former queries. Thus, sampling from a Gaussian process now relates to sampling from the *posterior* distribution. To do so, it is necessary to keep track and store former queries $x_1, \ldots, x_{i-1}$ and the privatized outputs $G_i = (\hat{g}_{DP}(x_1; \hat{\eta}_{DP}), \ldots, \hat{g}_{DP}(x_{i-1}; \hat{\eta}_{DP}))^T$.

---

**Algorithm 2:** Pseudo-code of DP-CATE for functions (iterative setting)

---

**Input:** CATE meta-learner $\hat{g}_D(\cdot; \hat{\eta}_{DP})$ trained on dataset $D$ with $|D| = n$, Gaussian kernel matrix conditioned on the former queries $x_1, \ldots, x_{i-1}$, former private outputs $G_i = (\hat{g}_{DP}(x_1; \hat{\eta}_{DP}), \ldots, \hat{g}_{DP}(x_{i-1}; \hat{\eta}_{DP}))^T$, new query $x_i$, privacy budget $\varepsilon$, $\delta$, Lipschitz const. $L$, ridge regularization $\lambda$

**Output:** Privatized new prediction $\hat{g}_{DP}(x_i; \hat{\eta}_{DP})$

**if** *i=1* **then**
```
  | /* Apply Algorithm 1                                             */
```
**end**
**else**
```
  | /* Calculate pairwise kernel vector                              */
```
  $\quad V_i \leftarrow (K(x_1, x_i), \ldots, K(x_{i-1}, x_i))^T;$
```
  | /* Sample from Gaussian process posterior                        */
```
  $\quad C_i \leftarrow (K(x_k, x_l))_{k,l=1}^{i-1};$
  $\quad s \sim \mathcal{N}(V_i^T C_i^{-1} G_i, K(x_i, x_i) - V_i^T C_i^{-1} V_i);$
```
  | /* Return private estimate                                       */
```
  $\quad \hat{g}_{DP}(x_i; \hat{\eta}_{DP}) \leftarrow s;$
**end**

---

**A note on complexity:** Algorithm 2 requires storing and iterative updating the outcome vector $G_i$ and the inverse matrix $C_i^{-1}$. The computational complexity of Alg. 2 will thus grow with an increasing number of queries. This poses a limitation of the above approach for settings with many iterative queries.

# D  THEORETICAL EVALUATION OF EXCESS RISK

Let $\hat{g}_{\mathrm{DP}}(\cdot; \hat{\eta}_{\mathrm{DP}})$ and $\hat{g}_{\mathrm{D}}(\cdot; \hat{\eta}_{\mathrm{DP}})$ denote the private and non-private CATE estimates, respectively. Also, let $g^*(\cdot; \hat{\eta}_{\mathrm{DP}})$ denote the second-stage population minimizer. We follow the existing literature (e.g., Bassily et al., 2014) and measure success of our method through the worst-case expected *excess empirical risk* with respect to dataset $D$, defined as

$$\xi_D(\hat{g}_{\mathrm{DP}}, \hat{\eta}_{\mathrm{DP}}) := \mathbb{E}_A\Big[\big\|\hat{g}_{\mathrm{DP}}(\cdot; \hat{\eta}_{\mathrm{DP}}) - g^*(\cdot; \hat{\eta}_{\mathrm{DP}})\big\|_{L_2}^2 - \big\|\hat{g}_D(\cdot; \hat{\eta}_{\mathrm{DP}}) - g^*(\cdot; \hat{\eta}_{\mathrm{DP}})\big\|_{L_2}^2\Big], \quad (23)$$

where the expectation is taken over the randomness in the second-stage privatization algorithm.

First, observe that, for a suitable calibration factor $r(\varepsilon, \delta, \hat{g}_D, \hat{\eta}_{\mathrm{DP}})$ and noise variable $U$ specified in Theorems 1 and 3, we can bound the general formula of the excess risk by

$$\xi_D(\hat{g}_{\mathrm{DP}}, \hat{\eta}_{\mathrm{DP}}) = \mathbb{E}_U\Big[\big\|\hat{g}_D(\cdot; \hat{\eta}_{\mathrm{DP}}) + r(\varepsilon, \delta, \hat{g}_D, \hat{\eta}_{\mathrm{DP}}) \cdot U - g^*(\cdot; \hat{\eta}_{\mathrm{DP}})\big\|_{L_2}^2 \qquad (24)$$

$$- \big\|\hat{g}_D(\cdot; \hat{\eta}_{\mathrm{DP}}) - g^*(\cdot; \hat{\eta}_{\mathrm{DP}})\big\|_{L_2}^2\Big]$$

$$\leq r(\varepsilon, \delta, \hat{g}_D, \hat{\eta}_{\mathrm{DP}})^2 \cdot \mathbb{E}_U[\|U\|_{L_2}^2] \qquad (25)$$

**Finitely many queries:** From Theorem 1, we have $U = \mathbf{U} \sim \mathcal{N}(0, \boldsymbol{I}_d)$ and, by setting $d = 1$, we yield

$$r(\varepsilon, \delta, \hat{g}_D, \hat{\eta}_{\mathrm{DP}}) = \gamma(T, D) \cdot \frac{5\sqrt{2\ln(n)\ln(2/\delta)}}{\varepsilon\, n}, \qquad (26)$$

where $n$ is the sample size of the training data $D$. As $\mathbb{E}_U[\|U\|_{L_2}^2] = \mathbb{E}_U[U^2] = 1$, the excess risk can be bounded by

$$\xi_D(\hat{g}_{\mathrm{DP}}, \hat{\eta}_{\mathrm{DP}}) \leq \gamma(T, D) \cdot \frac{50 d\ln(n)\ln(2/\delta)}{(\varepsilon\, n)^2} = O(n^{-2}\ln(n)) \in o(n^{-1}). \qquad (27)$$

**Functional queries:** From Theorem 3, we have

$$r(\varepsilon, \delta, \hat{g}_D, \hat{\eta}_{\mathrm{DP}}) = \sup_{(a,x)\in\{0,1\}\times\mathcal{X}} \big[\rho(a, \hat{\pi}_{\mathrm{DP}}(x))\big] \frac{4L\sqrt{2\ln(2/\delta)}}{(\sqrt{2\pi}h)^q \lambda n\varepsilon}, \qquad (28)$$

where $h$ denotes the kernel bandwidth and $q$ the dimension of the covariates. Instead of perturbing a random variable $U$ as above, Theorem 3 perturbs the functional output by a sample path $U = U(\cdot) \in \mathcal{H}$ of a zero-centered Gaussian process with covariance function $K(x, x') = (\sqrt{2\pi}h)^{-q}\exp(-\|x - x'\|_2^2/(2h^2))$. First, we note that

$$\mathbb{E}_U[\mathbb{E}_X(U(X)^2)] = \mathbb{E}_X[\mathbb{E}_U(U(X)^2)] = \mathbb{E}_X[K(X, X)] = (\sqrt{2\pi}h)^{-q}. \qquad (29)$$

Therefore, we get

$$\xi_D(\hat{g}_{\mathrm{DP}}, \hat{\eta}_{\mathrm{DP}}) \leq \Big(\sup_{(a,x)\in\{0,1\}\times\mathcal{X}} \big[\rho(a, \hat{\pi}_{\mathrm{DP}}(x))\big]\Big)^2 \frac{32L^2\ln(2/\delta)}{(\sqrt{2\pi}h)^{3q}(\lambda n\varepsilon)^2} = O(n^{-2}) \in o(n^{-1}).$$

$$(30)$$

# E    SUPPORTING LEMMAS AND DEFINITIONS

Our proofs are based on important properties of differentially private algorithms, which we introduce below. For proofs and further details, see (Avella-Medina, 2021; Dwork & Roth, 2014; Nissim et al., 2007).

**Lemma 2** (Post-processing theorem). *Let the mechanism $\mathcal{M}$ satisfy $(\varepsilon, \delta)$-DP. For any function $f$, it then holds that $f(\mathcal{M})$ satisfies $(\varepsilon, \delta)$-DP as well.*

**Lemma 3** (Sequential composition property). *Let the set of privacy mechanisms $\mathcal{M}_j$, $j = 1, \ldots, k$ satisfy $(\varepsilon_j, \delta_j)$-DP. When applying the mechanisms $\mathcal{M}_j$ on the same dataset, the resulting overall mechanism (i.e., the concatenation of all the $\mathcal{M}_j$) guarantees $(\sum_{j=1}^{k} \varepsilon_j, \sum_{j=1}^{k} \delta_j)$-DP.*

**Definition 5** ($\xi$-smooth and local sensitivities). *Let $\hat{f}_D$ be a learned function on dataset $D$ with $|D| = n$ and let $\mathbf{x} \in \mathcal{X}^d$. Then, local sensitivity of $\hat{f}$ is defined as*

$$LS(\hat{f}, D) := \sup_{D' \sim D, \, \mathbf{x} \in \mathcal{X}^d} \|\hat{f}_D(\mathbf{x}) - \hat{f}_{D'}(\mathbf{x})\|_2. \tag{31}$$

*where the supremum is taken wrt. $D'$ (unlike $\Delta_2$, where the supremum is taken wrt. to both $D$ and $D'$). Furthermore, a $\xi$-smooth sensitivity of $\hat{f}$ is defined as*

$$SS_\xi(\hat{f}, D) := \sup_{D' \in \mathcal{Z}^n} \left[ \exp(-\xi d_{\mathrm{H}}(D, D')) \, LS(\hat{f}, D') \right], \tag{32}$$

*where $\mathcal{Z}^n$ denotes the data domain.*

**Lemma 4** (Differential privacy via $\xi$-smooth sensitivity (Avella-Medina, 2021; Nissim et al., 2007)). *Let $\hat{f}_D$ be a learned function on dataset $D$ with $|D| = n$ and let $\mathbf{x} \in \mathcal{X}^d$. Then, $\hat{f}_{\mathrm{DP}}(\mathbf{x})$ fulfills $(\varepsilon, \delta)$-differential privacy:*

$$\hat{f}_{\mathrm{DP}}(\mathbf{x}) = \hat{f}_D(\mathbf{x}) + \frac{5\sqrt{2\log(2/\delta)}}{\varepsilon} SS_\xi(\hat{f}, D) \cdot \mathbf{U}, \tag{33}$$

*where $\mathbf{U} \sim \mathcal{N}(0, \boldsymbol{I}_d)$ and $\xi = \frac{\varepsilon}{4(d + 2\log(2/\delta))}$.*

Finally, we show that in our case, the $\xi$-smooth sensitivity can be upper-bounded by the (appropriately scaled) gross-error sensitivity.

**Lemma 5.** (Avella-Medina, 2021) *Let $\hat{f}_D$ be a learned function on dataset $D$ with $|D| = n$ and let $\mathbf{x} \in \mathcal{X}^d$. Furthermore, let $\gamma(T, D)$ denote the sample gross error sensitivity of the parameter of the interest $T(D) = \hat{f}_D(\mathbf{x})$. Under minimal regularity and boundedness conditions for the $\xi$-smooth sensitivity $SS_\xi(\hat{f}, D)$ with $\xi = \frac{\varepsilon}{4(d + 2\log(2/\delta))}$, it holds that*

$$SS_\xi(\hat{f}, D) \leq \frac{\sqrt{\log(n)}}{n} \gamma(T, D). \tag{34}$$

# F PROOFS

## F.1 PROOFS OF THEOREM 1, COROLLARIES 1, 2, THEOREM 2

**Theorem 1** (DP-CATE for finite queries). *Let $z := (a, x, y)$ define a data sample following the joint distribution $\mathcal{Z}$ and $\hat{\eta}_{\mathrm{DP}} = (\hat{\mu}_{\mathrm{DP}}, \hat{\pi}_{\mathrm{DP}})$ the nuisance functions estimated on dataset $\tilde{D}$ in a $(\varepsilon/2, \delta/2)$-differentially private manner of choice.[8] Furthermore, let $D$ be the training dataset for the second-stage model with $|D| = n$. For $z = (a, x, y) \in \mathcal{Z}$, we define*

$$\hat{g}_{\mathrm{DP}}(\mathbf{x}; \hat{\eta}_{\mathrm{DP}}) := \hat{g}_D(\mathbf{x}; \hat{\eta}_{\mathrm{DP}}) + \underbrace{\gamma(T, D) \cdot c(\varepsilon, \delta, n)}_{r(\varepsilon, \delta, \hat{g}_D, \hat{\eta}_{\mathrm{DP}})} \cdot \mathbf{U}, \qquad T(P) = g^*(\mathbf{x}; \hat{\eta}_{\mathrm{DP}}), \tag{10}$$

$$\gamma(T, P) = \sup_{z \in \mathcal{Z}} \left\| h(g^*, \mathbf{x}, x, \hat{\eta}_{\mathrm{DP}}) \rho(a, \hat{\pi}_{\mathrm{DP}}(x)) (\phi(z, \hat{\eta}_{\mathrm{DP}}, \lambda(\hat{\pi}_{\mathrm{DP}}(x))) - g^*(x; \hat{\eta}_{\mathrm{DP}})) \right\|_2, \tag{11}$$

*where $\gamma(T, D)$ is a sample gross-error sensitivity with $T(D)$ instead of $T(P)$ in Eq. (9), $\mathbf{U} \sim \mathcal{N}(0, \boldsymbol{I}_d)$, $c(\varepsilon, \delta, n) := 5\sqrt{2 \ln(n) \ln(2/\delta)}/(\varepsilon n)$, and where $h(g^*, \mathbf{x}, z, \hat{\eta}_{\mathrm{DP}}) \in \mathbb{R}^d$ and $g^*(\cdot; \hat{\eta}_{\mathrm{DP}})$ depend on the machine learning model employed for the second-stage regression. Then, $\hat{g}_{\mathrm{DP}}(\mathbf{x}; \hat{\eta}_{\mathrm{DP}})$ is $(\varepsilon, \delta)$-differentially private.*

*Proof.* First, observe that by Lemma 3, the first-stage nuisance estimation $\hat{\eta}_{\mathrm{DP}}$ overall fulfills $(\varepsilon, \delta)$-DP. By Lemma 2, the privacy of the nuisances is not affected by the second-stage regression. Note that we estimate both stages on disjoint subsets of the data, i.e., $\bar{D} = \tilde{D} \cup D$.

If we had access to the $\xi$-smooth sensitivity $SS_\xi(g, D)$ of the meta-learner with $\xi = \frac{\varepsilon}{4(d+2\log(2/\delta))}$, the estimator $\hat{g}_{\mathrm{DP}}(\mathbf{x}; \hat{\eta}_{\mathrm{DP}})$ with

$$\hat{g}_{\mathrm{DP}}(\mathbf{x}; \hat{\eta}_{\mathrm{DP}}) = \hat{g}_D(\mathbf{x}; \hat{\eta}_{\mathrm{DP}}) + \frac{5\sqrt{2\log(2/\delta)}}{\varepsilon} SS_\xi(\hat{g}, D) \cdot \mathbf{U}, \tag{35}$$

where $\mathbf{U} \sim \mathcal{N}(0, \boldsymbol{I}_d)$, $\mathbf{x} \in \mathbb{R}^d$, would fulfill $(\varepsilon, \delta)$-differential privacy by Lemma 4 and the parallel composition property of DP.

However, the $\xi$-smooth sensitivity is highly difficult, or even impossible, to compute for general function classes such as neural networks. Therefore, we seek an upper bound on $SS_\xi(g, D)$ to ensure that the privacy guarantees stay valid while making it feasible to compute the calibration function $r(\varepsilon, \delta, \hat{g}_D, \hat{\eta}_{\mathrm{DP}})$. By Lemma 5, we find such an upper bound through

$$SS_\xi(\hat{g}, D) \leq \frac{\sqrt{\log(n)}}{n} \gamma(T, D) = \frac{\sqrt{\log(n)}}{n} \sup_{z \in \mathcal{Z}} \|\mathrm{IF}(z, T; D)\|_2, \tag{36}$$

where $\gamma(T, D)$ denotes the sample gross-error sensitivity with the parameter of interest $T$ defined as $T(P) = g^*(\mathbf{x}; \hat{\eta}_{\mathrm{DP}})$ (population version) or $T(D) = \hat{g}_D(\mathbf{x}; \hat{\eta}_{\mathrm{DP}})$ (sample version).

Now, the influence function of $T(P) = g^*(\mathbf{x}; \hat{\eta}_{\mathrm{DP}})$ where $g^*(\cdot; \hat{\eta}_{\mathrm{DP}}) = \arg\min_{g \in \mathcal{G}} R_P(g, \hat{\eta}_{\mathrm{DP}}, \lambda(\hat{\pi}_{\mathrm{DP}}))$ is as follows:

$$\mathrm{IF}(z, T; P) = \frac{\mathrm{d}}{\mathrm{d}t} \left[ \left( \arg\min_{g \in \mathcal{G}} R_{(1-t)P + t\delta_z}(g, \hat{\eta}_{\mathrm{DP}}, \lambda(\hat{\pi}_{\mathrm{DP}})) \right)(\mathbf{x}) \right] \Bigg|_{t=0} \tag{37}$$

$$= \frac{\mathrm{d}}{\mathrm{d}t} \left[ \left( \arg\min_{g \in \mathcal{G}} \left\{ (1-t)R_P(g, \hat{\eta}_{\mathrm{DP}}, \lambda(\hat{\pi}_{\mathrm{DP}})) + t\rho(a, \hat{\pi}_{\mathrm{DP}}(x))(\phi(z, \hat{\eta}_{\mathrm{DP}}, \lambda(\hat{\pi}_{\mathrm{DP}}) - g(x))^2 \right\} \right)(\mathbf{x}) \right] \Bigg|_{t=0}. \tag{38}$$

Here, the $\arg\min$ is achieved at $g_t^*(\cdot; \hat{\eta}_{\mathrm{DP}})$, and, for smooth models $\mathcal{G}$, the following holds:

$$\frac{\mathrm{d}}{\mathrm{d}g} \left\{ (1-t)R_P(g, \hat{\eta}_{\mathrm{DP}}, \lambda(\hat{\pi}_{\mathrm{DP}})) + t\rho(a, \hat{\pi}_{\mathrm{DP}}(x))(\phi(z, \hat{\eta}_{\mathrm{DP}}, \lambda(\hat{\pi}_{\mathrm{DP}}) - g(x))^2 \right\} \Bigg|_{g = g_t^*(\cdot; \hat{\eta}_{\mathrm{DP}})} = 0, \tag{39}$$

if and only if

$$\underbrace{(1-t)\frac{\mathrm{d}}{\mathrm{d}g} \left\{ R_P(g, \hat{\eta}_{\mathrm{DP}}, \lambda(\hat{\pi}_{\mathrm{DP}})) \right\} - 2t\rho(a, \hat{\pi}_{\mathrm{DP}}(x))(\phi(z, \hat{\eta}_{\mathrm{DP}}, \lambda(\hat{\pi}_{\mathrm{DP}})) - g(x))\frac{\mathrm{d}g(x)}{\mathrm{d}g}}_{F(g,t)} \Bigg|_{g = g_t^*(\cdot; \hat{\eta}_{\mathrm{DP}})} = 0.$$

$$\tag{40}$$

---

[8]As we need to query the nuisance functions multiple times, the privatization method needs to be suitable for functions, such as gradient perturbation through DP-SGD (Abadi et al., 2016).

Therefore, by the implicit function theorem:

$$\frac{\mathrm{d}}{\mathrm{d}t}\Big[g_t^*(\mathbf{x};\hat{\eta}_{\mathrm{DP}})\Big]\Big|_{t=0} = \left(\frac{\mathrm{d}F(g,t)}{\mathrm{d}g}\Big|_{t=0;\, g=g^*(\mathbf{x};\hat{\eta}_{\mathrm{DP}})}\right)^{-1}\frac{\mathrm{d}F(g,t)}{\mathrm{d}t}\Big|_{t=0;\, g=g^*(\mathbf{x};\hat{\eta}_{\mathrm{DP}})} \tag{41}$$

$$= \left(\frac{\mathrm{d}^2}{\mathrm{d}^2 g}\big\{R_P(g,\hat{\eta}_{\mathrm{DP}},\lambda(\hat{\pi}_{\mathrm{DP}}))\big\}\Big|_{g=g^*(\mathbf{x};\hat{\eta}_{\mathrm{DP}})}\right)^{-1}\Big(\underbrace{-\frac{\mathrm{d}}{\mathrm{d}g}\big\{R_P(g,\hat{\eta}_{\mathrm{DP}},\lambda(\hat{\pi}_{\mathrm{DP}}))\big\}\Big|_{g=g^*(\mathbf{x};\hat{\eta}_{\mathrm{DP}})}}_{=0}$$

$$- 2\rho(a,\hat{\pi}_{\mathrm{DP}}(x))(\phi(z,\hat{\eta}_{\mathrm{DP}},\lambda(\hat{\pi}_{\mathrm{DP}})) - g^*(x;\hat{\eta}_{\mathrm{DP}})) \cdot \frac{\mathrm{d}g(x)}{\mathrm{d}g}\Big|_{g=g^*(x;\hat{\eta}_{\mathrm{DP}})}\Big). \tag{42}$$

Now, by regrouping and setting $h(g^*,\mathbf{x},x,\hat{\eta}_{\mathrm{DP}})$ as

$$h(g^*,\mathbf{x},x,\hat{\eta}_{\mathrm{DP}}) = 2\left(\frac{\mathrm{d}^2}{\mathrm{d}^2 g}\big\{R_P(g,\hat{\eta}_{\mathrm{DP}},\lambda(\hat{\pi}_{\mathrm{DP}}))\big\}\Big|_{g=g^*(\mathbf{x};\hat{\eta}_{\mathrm{DP}})}\right)^{-1} \cdot \frac{\mathrm{d}g(x)}{\mathrm{d}g}\Big|_{g=g^*(x;\hat{\eta}_{\mathrm{DP}})}, \tag{43}$$

we obtain the desired influence function:

$$\mathrm{IF}(z,T;P) = -h(g^*,\mathbf{x},x,\eta) \cdot \rho(a,\hat{\pi}_{\mathrm{DP}}(x))(\phi(z,\hat{\eta}_{\mathrm{DP}},\lambda(\hat{\pi}_{\mathrm{DP}})) - g^*(x;\hat{\eta}_{\mathrm{DP}})). \tag{44}$$

Here, $h(g^*,\mathbf{x},x,\hat{\eta}_{\mathrm{DP}}) \in \mathbb{R}^d$ depends on the machine learning model employed for the second-stage regression (see Corollaries 1 and 2). Finally, taking the supremum over the data space $\mathcal{Z}$ and evaluating $\mathrm{IF}(z,T,P)$ at the empirical distribution $D$ with trained functions $\hat{\eta}_{\mathrm{DP}}$ and $\hat{g}_D(\cdot;\hat{\eta}_{\mathrm{DP}})$ states the desired result. □

**Corollary 1** (Parametric second-stage regression)**.** *If the second-stage regression is a smooth parametric model, namely $\mathcal{G} = \{g(x;\theta) : \mathcal{X} \mapsto \mathbb{R}, \theta \in \Theta \subseteq \mathbb{R}^p\}$, then, in Theorem 1, we have*

$$g^*(\cdot;\hat{\eta}_{\mathrm{DP}}) = g(\cdot;\theta^*) \quad and \quad h(g^*,\mathbf{x},x,\hat{\eta}_{\mathrm{DP}}) = 2\, J_\theta\big[g^*(\mathbf{x};\hat{\eta}_{\mathrm{DP}})\big] \cdot H_\theta^{-1} \cdot \nabla_\theta\big[g^*(x;\hat{\eta}_{\mathrm{DP}})\big], \tag{13}$$

*where $\theta^* = \arg\min\limits_{\theta\in\Theta} R_P(g,\hat{\eta}_{\mathrm{DP}},\lambda(\hat{\pi}_{\mathrm{DP}}))$; $J_\theta \in \mathbb{R}^{d\times p}$ is a Jacobian matrix wrt. $\theta$; $H_\theta = \nabla_\theta^2\big[R_P(g^*,\hat{\eta}_{\mathrm{DP}},\lambda(\hat{\pi}_{\mathrm{DP}}))\big] \in \mathbb{R}^{p\times p}$ is a Hessian matrix; and $\nabla_\theta$ is a gradient.*

*Proof.* First, we note, that $\theta^*$ is an example of the M-estimator. Therefore, $\mathrm{IF}(z,T;P)$ yields an influence function of the M-estimator. Specifically, it can be shown that

$$\left(\frac{\mathrm{d}^2}{\mathrm{d}^2 g}\big\{R_P(g,\hat{\eta}_{\mathrm{DP}},\lambda(\hat{\pi}_{\mathrm{DP}}))\big\}\Big|_{g=g^*(\mathbf{x};\hat{\eta}_{\mathrm{DP}})}\right)^{-1} = J_\theta\big[g^*(\mathbf{x};\hat{\eta}_{\mathrm{DP}})\big] \cdot H_\theta^{-1} \text{ and} \tag{45}$$

$$\frac{\mathrm{d}g(x)}{\mathrm{d}g}\Big|_{g=g^*(x;\hat{\eta}_{\mathrm{DP}})} = \nabla_\theta\big[g^*(x;\hat{\eta}_{\mathrm{DP}})\big]. \tag{46}$$

Therefore, we yield the desired result

$$h(g^*,\mathbf{x},x,\hat{\eta}_{\mathrm{DP}}) = 2\, J_\theta\big[g^*(\mathbf{x};\hat{\eta}_{\mathrm{DP}})\big] \cdot H_\theta^{-1} \cdot \nabla_\theta\big[g^*(x;\hat{\eta}_{\mathrm{DP}})\big]. \tag{47}$$

□

**Remark 1.** *Importantly, for the sample gross error sensitivity, $\gamma(T,D) = \sup_{z\in\mathcal{Z}}\|\mathrm{IF}(z,T;D)\|_2$, to be finite, we need additional regularity conditions. Specifically, we denote the score function of Eq. (3) by*

$$\psi(z,g,\hat{\eta}_{\mathrm{DP}}) := \rho(a,\pi(x))\,(\phi(z,\hat{\eta}_{\mathrm{DP}},\lambda(\hat{\pi}_{\mathrm{DP}})) - g(x)). \tag{48}$$

*First, note that $\psi(z,g,\hat{\eta}_{\mathrm{DP}})$ is differentiable w.r.t. $g(x)$ for all $z \in \mathcal{Z}$. Since we assume that our data stems from a bounded domain, there exist constants $K$ and $L$ such that $\psi(z,g,\hat{\eta}_{\mathrm{DP}})$ and the derivative $\psi'(z,g,\hat{\eta}_{\mathrm{DP}})$ are uniformly bounded in $\mathcal{Z}$ by $K$ and $L$. Additionally, we assume that $\|\nabla_\theta\hat{g}_D(x;\hat{\eta}_{\mathrm{DP}})\|_2 \leq \tilde{K}$. The constant $\tilde{K}$ can be regulated through, e.g., gradient clipping. Furthermore, the Hessian $H_\theta$ of the loss w.r.t. $\theta$ is positive semi-definite by assumption.*

*If $H_\theta$ is further restricted to be positive definite, we can employ Lemmas 1 and 2 from Avella-Medina (2021) to show the desired upper bound of the smooth sensitivity for a sufficiently large sample size*

$n$.[9] *Specifically, the minimum sample size depends inversely on the privacy budget $\varepsilon$ and $\delta$ and grows in the bounds $L, K, \tilde{K}$ and the ratio of the maximum and minimum eigenvalue of $H_\theta$.*

*Note: If the Hessian does not fulfill the above criteria and has negative eigenvalues, one can employ the "damping" trick (Martens, 2010) by approximating $H_\theta$ by $\tilde{H}_\theta := H_\theta + \alpha \mathbf{I}$ for $\alpha > 0$. The parameter $\alpha$ defines the conservativeness of the approximation.*

**Corollary 2** (Non-parametric second-stage regression). *If the second-stage regression is a kernel ridge regression with $\Lambda(g) = \lambda\|g\|_\mathcal{H}^2$, where $\mathcal{G} = \mathcal{H}$ is a reproducing kernel Hilbert space (RKHS) induced by a kernel $K(\cdot, \cdot) : \mathcal{X} \times \mathcal{X} \mapsto \mathbb{R}^+$, then, in Theorem 1, we have*

$$g^*(\cdot; \hat{\eta}_{\mathrm{DP}}) = \left(\mathbb{L}_\rho + \lambda\mathbb{I}\right)^{-1} S(\cdot) \quad and \quad h(g^*, \mathbf{x}, x, \hat{\eta}_{\mathrm{DP}}) = \left(\mathbb{L}_\rho + \lambda\mathbb{I}\right)^{-1} K(\cdot, x)(\mathbf{x}), \qquad (14)$$

*where $\mathbb{L}_\rho : \mathcal{H} \mapsto \mathcal{H}$ is a weighted covariance operator $\mathbb{L}_\rho f(\cdot) = \mathbb{E}\big[\rho(A, \hat{\pi}_{\mathrm{DP}}(X)) K(\cdot, X) f(X)\big]$; $\lambda\mathbb{I}f(\cdot) = \lambda f(\cdot)$ is a scaling operator; $S \in \mathcal{H}$ is a cross-covariance functional $S(\cdot) = \mathbb{E}\big[\rho(A, \hat{\pi}_{\mathrm{DP}}(X)) K(\cdot, X) \phi(Z, \hat{\eta}_{\mathrm{DP}}, \lambda(\hat{\pi}_{\mathrm{DP}}(X)))\big]$.*

*Proof.* By the representer theorem (population version), the optimal solution of the weighted kernel ridge regression is given by $g^*(\cdot; \hat{\eta}_{\mathrm{DP}}) = \left(\mathbb{L}_\rho + \lambda\mathbb{I}\right)^{-1} S(\cdot)$, where $\mathbb{L}_\rho f(\cdot) = \mathbb{E}\big[\rho(A, \hat{\pi}_{\mathrm{DP}}(X)) K(\cdot, X) f(X)\big]$ and $S(\cdot) = \mathbb{E}\big[\rho(A, \hat{\pi}_{\mathrm{DP}}(X)) K(\cdot, X) \phi(Z, \hat{\eta}_{\mathrm{DP}}, \lambda(\hat{\pi}_{\mathrm{DP}}))\big]$. For a description, see, e.g., Berlinet & Thomas-Agnan (2004); Pillonetto et al. (2022).

We now turn to deriving the influence function of $g^*(\mathbf{x}; \hat{\eta}_{\mathrm{DP}})$ at $z = (a, x, y)$, $\mathrm{IF}(z, T = g^*(\mathbf{x}; \hat{\eta}_{\mathrm{DP}}); P)$, to then receive the explicit form of $h(g^*, \mathbf{x}, x, \hat{\eta}_{\mathrm{DP}})$.

Consider the point-mass perturbation of the distribution $P$ in the direction of $z$ denoted as $P_t = (1 - t)P + t\delta_z$. Then, the solution to the kernel ridge regression at $P_t$ is given by

$$g_t^*(\mathbf{x}; \hat{\eta}_{\mathrm{DP}}) = \left(\mathbb{L}_{P_t} + \lambda\mathbb{I}\right)^{-1} S_{P_t}(\mathbf{x}), \qquad (49)$$

where $\mathbb{L}_{P_t} = (1 - t)L_\rho + tL_{\delta_z}$ and $S_{P_t} = (1 - t)S + tS_{\delta_z}$ with $L_{\delta_z} f(\mathbf{x}) = \rho(a, \hat{\pi}_{\mathrm{DP}}(x)) K(\mathbf{x}, x) f(z)$ and $S_{\delta_z}(\mathbf{x}) = \rho(a, \hat{\pi}_{\mathrm{DP}}(x)) K(\mathbf{x}, x) \phi(z, \hat{\eta}_{\mathrm{DP}}, \lambda(\hat{\pi}_{\mathrm{DP}}))$.

Then we get

$$\mathrm{IF}(z, T = g^*(\mathbf{x}; \hat{\eta}_{\mathrm{DP}}); P) = \frac{\mathrm{d}}{\mathrm{d}t} g_t^*(\mathbf{x}; \hat{\eta}_{\mathrm{DP}})\Big|_{t=0} \qquad (50)$$

$$= -\left(\mathbb{L}_\rho + \lambda\mathbb{I}\right)^{-1} L_{\delta_z}(\mathbf{x}) \left(\mathbb{L}_\rho + \lambda\mathbb{I}\right)^{-1} S(\mathbf{x}) + \left(\mathbb{L}_\rho + \lambda\mathbb{I}\right)^{-1} S_{\delta_z}(\mathbf{x}) \qquad (51)$$

$$= -\rho(a, \hat{\pi}_{\mathrm{DP}}(x)) \cdot \left(\left(\mathbb{L}_\rho + \lambda\mathbb{I}\right)^{-1} K(\cdot, x)\right)(\mathbf{x}) \cdot g^*(x; \hat{\eta}_{\mathrm{DP}}) \qquad (52)$$

$$+ \rho(a, \hat{\pi}_{\mathrm{DP}}(x)) \cdot \left(\left(\mathbb{L}_\rho + \lambda\mathbb{I}\right)^{-1} K(\cdot, x)\right)(\mathbf{x}) \cdot \phi(z, \hat{\eta}_{\mathrm{DP}}, \lambda(\hat{\pi}_{\mathrm{DP}})). \qquad (53)$$

With $h(g^*, \mathbf{x}, x, \hat{\eta}_{\mathrm{DP}}) = \left(\left(\mathbb{L}_\rho + \lambda\mathbb{I}\right)^{-1} K(\cdot, x)\right)(\mathbf{x})$ we get

$$\mathrm{IF}(z, g^*(\mathbf{x}; \hat{\eta}_{\mathrm{DP}}); P) = h(g^*, \mathbf{x}, x, \hat{\eta}_{\mathrm{DP}}) \cdot \rho(a, \hat{\pi}_{\mathrm{DP}}) \big(\phi(z, \hat{\eta}_{\mathrm{DP}}, \lambda(\hat{\pi}_{\mathrm{DP}})) - g^*(x; \hat{\eta}_{\mathrm{DP}})\big), \quad (54)$$

which shows the desired result.

$\square$

**Theorem 2** (Neyman-orthogonality and quasi-oracle efficiency of DP-CATE). *The privatization of the second-stage model asymptotically preserves the property of Neyman-orthogonality, namely*

$$\|g^*(\cdot; \eta) - \hat{g}_{\mathrm{DP}}(\cdot; \hat{\eta}_{\mathrm{DP}})\|_{L_2}^2 \lesssim R_P\big(g^*(\cdot; \hat{\eta}_{\mathrm{DP}}), \hat{\eta}_{\mathrm{DP}}, \lambda(\hat{\pi}_{\mathrm{DP}})\big) - R_P\big(g^*(\cdot; \eta), \hat{\eta}_{\mathrm{DP}}, \lambda(\hat{\pi}_{\mathrm{DP}})\big) + R_2(\hat{\eta}_{\mathrm{DP}}, \eta)$$
$$+ \underbrace{\|g^*(\cdot; \hat{\eta}_{\mathrm{DP}}) - \hat{g}_D(\cdot; \hat{\eta}_{\mathrm{DP}})\|_{L_2}^2}_{depends\ on\ the\ model\ class\ \mathcal{G}} + \underbrace{o_P(n^{-1})}_{output\ perturbation}. \qquad (15)$$

*Furthermore, under additional regularity conditions on the privatization of the nuisance functions (e.g., gradient perturbation), our DP-CATE achieves quasi-oracle efficiency. Specifically, if the original estimation of the nuisance functions is at rate of at least $o_P(n^{-1/4})$, then the privatized estimation preserves this rate.*

---

[9]For details, we refer to Avella-Medina (2021).

*Proof.* First, we show that the privatization at the second stage of learning, namely the output perturbation, asymptotically does not affect the Neyman-orthogonality. Specifically, we show that the $L_2$ error can be upper-bounded as

$$\|g^*(\cdot;\eta) - \hat{g}_{\text{DP}}(\cdot;\hat{\eta}_{\text{DP}})\|_{L_2}^2 \tag{55}$$

$$= \|g^*(\cdot;\eta) - \hat{g}_D(\cdot;\hat{\eta}_{\text{DP}}) - r(\varepsilon,\delta,\hat{g}_D,\hat{\eta}_{\text{DP}}) \cdot U\|_{L_2}^2 \tag{56}$$

$$\leq \|g^*(\cdot;\eta) - \hat{g}_D(\cdot;\hat{\eta}_{\text{DP}})\|_{L_2}^2 + r(\varepsilon,\delta,\hat{g}_D,\hat{\eta}_{\text{DP}}) \cdot \|U\|_{L_2}^2 \tag{57}$$

$$\leq \|g^*(\cdot;\eta) - g^*(\cdot;\hat{\eta}_{\text{DP}})\|_{L_2}^2 + \underbrace{\|g^*(\cdot;\hat{\eta}_{\text{DP}}) - \hat{g}_D(\cdot;\hat{\eta}_{\text{DP}})\|_{L_2}^2}_{\text{depends on the model class }\mathcal{G}} + \underbrace{o_P(n^{-1})}_{\text{output perturbation}}, \tag{58}$$

where the last inequality holds due to the excess risk of the output perturbation (see Appendix D). Now, we can also employ the result from Morzywolek et al. (2023):

$$\|g^*(\cdot;\eta) - g^*(\cdot;\hat{\eta}_{\text{DP}})\|_{L_2}^2 \lesssim R_P\big(g^*(\cdot;\hat{\eta}_{\text{DP}}),\hat{\eta}_{\text{DP}},\lambda(\hat{\pi}_{\text{DP}})\big) - R_P\big(g^*(\cdot;\eta),\hat{\eta}_{\text{DP}},\lambda(\hat{\pi}_{\text{DP}})\big) + R_2(\hat{\eta}_{\text{DP}},\eta), \tag{59}$$

where $R_2(\hat{\eta}_{\text{DP}},\eta)$ is the higher-order error of misspecifying the nuisance functions. Specifically, the $R_2(\hat{\eta}_{\text{DP}},\eta)$ takes a different form for different learners:

$$\text{DR-learner:} \quad R_2(\hat{\eta}_{\text{DP}},\eta) = \sum_{a\in\{0,1\}} \|\mu(\cdot,a) - \hat{\mu}_{\text{DP}}(\cdot,a)\|_{L_2}^2 \cdot \|\pi - \hat{\pi}_{\text{DP}}\|_{L_2}^2; \tag{60}$$

$$\text{R-learner:} \quad R_2(\hat{\eta}_{\text{DP}},\eta) = \sum_{a\in\{0,1\}} \|\mu(\cdot,a) - \hat{\mu}_{\text{DP}}(\cdot,a)\|_{L_2}^2 \cdot \|\pi - \hat{\pi}_{\text{DP}}\|_{L_2}^2 + \|\pi - \hat{\pi}_{\text{DP}}\|_{L_4}^4. \tag{61}$$

Second, we want to demonstrate when the quasi-oracle efficiency holds for our DP-CATE. Specifically, given that the non-private estimators $\hat{\mu}_{\tilde{D}}$ and $\hat{\pi}_{\tilde{D}}$ are estimated at the rate of at least $o_P(n^{-1/4})$, we need to show when the privatized nuisance functions $\hat{\eta}_{\text{DP}}$ can also achive this rate.

We focus on the smooth parametric models for $\hat{\eta}_{\tilde{D}}$ and a basic gradient perturbation method for privatization, namely DP-SGD (Bassily et al., 2014; Abadi et al., 2016). Let $m$ denote the number of model parameters. From the literature, (e.g., Chen et al., 2020), we know that the convergence rate $r$ for $(\varepsilon,\delta)$-DP-SGD for convex and Lipschitz losses holds

$$r^{\text{DP-SGD}} = O_P\left(\frac{\sqrt{m}}{n\varepsilon}\right), \tag{62}$$

where $n$ denotes the sample size.

Recall that in our setup we only allocate the budget $(\varepsilon/2,\delta/2)$ to the privatization of each $\hat{\pi}_{\tilde{D}}$ and $\hat{\mu}_{\tilde{D}}$. We yield the following result by straightforward calculation: For $\hat{\pi}$ and $\hat{\mu}$ with $m \leq \sqrt{n}^3\varepsilon^2$, $r^{\text{DP-SGD}} \in o_P(n^{-1/4})$.

For other losses (e.g., non-Lipschitz, non-convex) as well as advanced versions of DP-SGD, an upper bound of $O_P\left(\frac{1}{\sqrt{n}} + \frac{\sqrt{m\log(1/\delta)}}{n\varepsilon}\right)$ has been derived in the literature. For an overview, see (Wang et al., 2024). In this case, we achieve

$$r^{\text{DP-SGD}} \in o_P(n^{-1/4}) \iff m\log(1/\delta) \leq n\varepsilon^2(n^{1/4}-1)^2. \tag{63}$$

Overall, we thus face a trade-off between the model size in terms of the number of parameters, privacy budget, and sample size. For appropriately chosen factors, the first-stage nuisances achieve rate $o_P(n^{-1/4})$, rendering DP-CATE oracle-efficient.

$\square$

### F.2 PROOFS OF LEMMA 1, THEOREM 3

**Lemma 1.** *Let $\mathcal{H}$ denote the RKHS induced by the Gaussian kernel $K(x,x') = (\sqrt{2\pi}h)^{-q}\exp(-\|x-x'\|_2^2/(2h^2))$ for $x,x' \in \mathcal{X} \subseteq \mathbb{R}^q$, and let $\hat{f}_D$ be the optimal solution to the RKHS regression*

$$\hat{f}_D(\cdot) = \arg\min_{f \in \mathcal{H}} \frac{1}{n} \sum_{i=1}^{n} w(X_i) \cdot \ell(f(X_i), Y_i) + \lambda \|f\|_{\mathcal{H}}^2, \tag{16}$$

*where $w(\cdot) > 0$ is a weight function, $D$ is a dataset with $|D| = n$, and $\ell(\hat{y}, y)$ is a convex and Lipschitz loss function in $\hat{y}$ with Lipschitz constant $L$. Then, for $D \sim D'$, we have*

$$\|\hat{f}_D - \hat{f}_{D'}\|_{\mathcal{H}} \leq \sup_{x \in \mathcal{X}} [w(x)] \frac{L}{\lambda n} \left( \sqrt{(2\pi)} h \right)^{-q}. \tag{17}$$

*Proof.* Observe that, for all $x \in \mathbb{R}^q$, the Gaussian kernel norm is given by $K(x, x) = \frac{1}{(\sqrt{2\pi}h)^q}$. Since the loss $l$ is convex and Lipschitz with constant $L$, the $w(\cdot)$-weighted loss is $\left(\sup_{x \in \mathcal{X}} [w(x)] \cdot L\right)$-Lipschitz. Thus, the overall loss in Eq. (16) is $\left(\sup_{x \in \mathcal{X}} [w(x)] \cdot L\right)$-admissable (see Hall et al., 2013). Therefore, we can employ a result from Hall et al. (2013), stating that the RKHS norm of minimizers of neighboring datasets can be bounded as

$$\|\hat{f}_D - \hat{f}_{D'}\|_{\mathcal{H}} \leq \sup_{x \in \mathcal{X}} [w(x)] \cdot \frac{L}{\lambda n} \sqrt{\sup_x K(x, x)}. \tag{64}$$

With our observation above, the result follows. $\qquad\square$

**Theorem 3** (DP-CATE for functional queries). *Let $\hat{\mu}_{\mathrm{DP}}$ and $\hat{\pi}_{\mathrm{DP}}$ denote the $(\varepsilon/2, \delta/2)$-differentially private nuisance estimators trained in stage 1 on $\tilde{D}$. Let $z = (a, x, y)$ be a data sample from dataset $D$ with $|D| = n$ and $x \in \mathcal{X} \subseteq \mathbb{R}^q$. Let $\mathcal{H}$ denote the RKHS induced by the kernel $K(x, x') = (\sqrt{2\pi}h)^{-q} \exp(-\|x - x'\|_2^2 / 2h^2)$, and let $\ell(\cdot, \cdot)$ be a convex and Lipschitz loss function with Lipschitz constant $L$. We define $\hat{g}_D(\cdot; \hat{\eta}_{\mathrm{DP}})$ as the second-stage regression solving Eq. (3) via*

$$\hat{g}_D(\cdot; \hat{\eta}_{\mathrm{DP}}) = \arg\min_{g \in \mathcal{H}} \frac{1}{n} \sum_{i=1}^{n} \rho(A_i, \hat{\pi}_{\mathrm{DP}}(X_i)) \ell(g(X_i), \phi(Z_i, \hat{\eta}_{\mathrm{DP}}, \lambda(\hat{\pi}_{\mathrm{DP}}(X_i)))) + \lambda \|g\|_{\mathcal{H}}^2. \tag{18}$$

*Furthermore, let $U(\cdot) \in \mathcal{H}$ be the sample path of a zero-centered Gaussian process with covariance function $K(x, x')$. Then, $(\varepsilon, \delta)$-differential privacy is guaranteed by*

$$\hat{g}_{\mathrm{DP}}(\cdot; \hat{\eta}_{\mathrm{DP}}) := \hat{g}_D(\cdot; \hat{\eta}_{\mathrm{DP}}) + \underbrace{\sup_{(a,x) \in \{0,1\} \times \mathcal{X}} [\rho(a, \hat{\pi}_{\mathrm{DP}}(x))] \frac{4L\sqrt{2\ln(2/\delta)}}{(\sqrt{2\pi}h)^q \lambda n \varepsilon}}_{r(\varepsilon, \delta, \hat{g}_D, \hat{\eta}_{\mathrm{DP}})} \cdot U(\cdot). \tag{19}$$

*Proof.* Let $U(\cdot)$ be the sample path of a zero-centered Gaussian process with covariance function $K(x, x')$. For our proof, we make use of Corollary 9 from Hall et al. (2013): for $\hat{f} \in \mathcal{H}$, where $\mathcal{H}$ is the RKHS corresponding to the kernel $K$, the release of

$$\tilde{f}_D(\cdot) = \hat{f}_D(\cdot) + \frac{\Delta \cdot c(\delta)}{\varepsilon} \cdot U(\cdot) \tag{65}$$

is $(\varepsilon, \delta)$-differentially private for

$$c(\delta) \geq \sqrt{2\log(\frac{2}{\delta})} \tag{66}$$

and

$$\Delta \geq \sup_{D \sim D'} \|\hat{f}_D - \hat{f}_{D'}\|_{\mathcal{H}}. \tag{67}$$

Therefore, for $\Delta_{\mathcal{H}} := \sup_{D \sim D'} \|\hat{g}_D(\cdot; \hat{\eta}_{\mathrm{DP}}) - \hat{g}_{D'}(\cdot; \hat{\eta}_{\mathrm{DP}})\|_{\mathcal{H}}$,

$$\hat{g}_{\mathrm{DP}}(\cdot; \hat{\eta}_{\mathrm{DP}}) = \hat{g}_D(\cdot; \hat{\eta}_{\mathrm{DP}}) + \frac{\Delta_{\mathcal{H}} \sqrt{2\log(2/\delta)}}{\varepsilon} \cdot U(\cdot) \tag{68}$$

is $(\varepsilon, \delta)$-differentially private. Note that here, again, we implicitly make use of Lemma 2 and Lemma 3 in the same manner as in the proof of Theorem 1.

Finally, from Lemma 1, we know that

$$\sup_{D \sim D'} \|\hat{g}_D(\cdot; \hat{\eta}_{\mathrm{DP}}) - \hat{g}_{D'}(\cdot; \hat{\eta}_{\mathrm{DP}})\|_{\mathcal{H}} \leq \frac{L}{\lambda n} (\sqrt{(2\pi)} h)^{-q}. \tag{69}$$

Thus, the desired result follows. $\qquad\square$

# G  EXPERIMENTS

## G.1  SYNTHETIC DATASET GENERATION

We consider two different data-generation settings with different complexity. Both settings follow the mechanism described in (Oprescu et al., 2019):

$$X_i \sim \mathcal{U}[0,1]^p, \tag{70}$$

$$A_i = \mathbf{1}\{(X^T\beta)_i \geq \eta_i\}, \tag{71}$$

$$Y_i = \theta(X_i)A_i + (X^T\gamma)_i + \epsilon_i, \tag{72}$$

where $\eta_i, \epsilon_i \sim \mathcal{U}[-1,1]$ and $\beta, \gamma$ have support with values drawn from $\mathcal{U}[0, 0.3]$ and $\mathcal{U}[0,1]$. The dimension of the covariates is set to $p = 2$ for Dataset 1 and $p = 30$ for Dataset 2. In Dataset 1, the conditional treatment effect $\theta(x)$ is defined as

$$\theta(x) = \exp(2x_0) + 3\sin(4x_0). \tag{73}$$

In Dataset 2, $\theta(x)$ is defined as

$$\theta(x) = \exp(2x_0) + 3\sin(4x_1). \tag{74}$$

For each setting, we draw 3000 samples, which we split into train (90%) and test (10%) sets.

## G.2  MEDICAL DATASETS

**MIMIC-III:** We showcase DP-CATE on the MIMIC-III dataset (Johnson et al., 2016), which includes electronic health records (EHRs) from patients admitted to intensive care units. We extract 8 confounders (heart rate, sodium, red blood cell count, glucose, hematocrit, respiratory rate, age, gender) and a binary treatment (mechanical ventilation) using an open-source preprocessing pipeline (Wang et al., 2020). We define the outcome variable as the red blood cell count after treatment which we adapt to be more responsive to the treatment ventilation. To extract features from the patient trajectories in the EHRs, we sample random time points and average the value of each variable over the ten hours prior to the sampled time point. All samples with missing values and outliers are removed from the dataset. We define samples with values smaller than the 0.1st percentile or larger than the 99.9th percentile of the corresponding variable as outliers. Our final dataset contains 14719 samples, which we split into train (90%) and test (10%) sets.

**TCGA:** The Cancer Genome Atlas (TCGA) dataset (Weinstein et al., 2013) contains a comprehensive and diverse collection of gene expression data collected from patients with different types of cancer. We consider the gene expression measurements of the 4,000 genes with the highest variability which we employ as our features $X$. The study cohort of consisted of 9659 patients. We model the binary treatment based on the sum of the 10 covariates with the highest variance and assign a constant treatment effect in the sum of the covariates.

## G.3  IMPLEMENTATION DETAILS

Our experiments are implemented in Python. We provide our code in our GitHub repository: `https://github.com/m-schroder/DP-CATE`.

Our DP-CATE is model-agnostic and highly flexible. Therefore, we implement multiple versions of the R- and the DR-leaner with varying base learner instantiations. For the outcome and the propensity estimation, we always employ a multilayer perceptron regression and classification model, respectively. The models consist of one layer of width 32 with ReLu activation function and were optimized via Adam at a learning rate of 0.01 and batch size 128.

For our experiments with the finite-query DP-CATE, we implement the pseudo-outcome regression in the second stage as (a) a kernel ridge regression model with a Gaussian kernel and default parameter specifications (KR) and (b) a neural network (NN) with two hidden layers of width 32 with tanh activation function trained in the same manner as the nuisance models. In the experiments for our functional DP-CATE, we employ a Gaussian kernel ridge regression with $m = 50$ basis functions and default regularization parameter $\lambda = 1$. Furthermore, our functional DP-CATE requires the specification of the Lipschitz constant $L$. This constant is either known based on the employed

loss, e.g., $L = 1$ for the $l_1$ loss or can be upper-bounded. In our settings, we employ the $l_2$ loss on a bounded domain. Therefore, although the $l_2$ loss itself is not Lipschitz, we can calculate $L$ numerically as the upper bound of the domain. We did not perform hyperparameter optimization, as our model-agnostic framework is applicable to any prediction model.

Our framework requires calculating the supremum of the influence functions. We implemented the maximization problem through mathematical optimization using the L-BFGS-B, a limited-memory Broyden–Fletcher–Goldfarb–Shanno algorithm for solving nonlinear optimization problems with bounded variables. The solver was run with default parameters.

