# OpenReview forum: "Differentially private learners for heterogeneous treatment effects"
_ICLR.cc/2025/Conference — ICLR 2025 Poster_

### Official Review · Reviewer_qbNM · 2024-10-31

**Soundness:** 3
**Presentation:** 4
**Contribution:** 2
**Rating:** 6
**Confidence:** 3

**Summary:**

The paper introduces a novel, privacy-preserving approach for estimating the Conditional Average Treatment Effect (CATE), motivated by the need for privacy in electronic health records. The authors propose DP-CATE, a flexible framework that ensures differential privacy while maintaining double robustness in CATE estimation. The framework is offered in two versions: one for finite queries (e.g., treatment effects for specific patient groups) and another for functional queries (releasing the complete CATE function). A key technical innovation lies in calibrating noise using influence functions for finite queries and Gaussian processes for functional queries. The authors provide theoretical privacy guarantees and demonstrate the framework's effectiveness using both synthetic data and real-world medical datasets (MIMIC-III and TCGA).

**Strengths:**

- The presentation of this paper is clear and well-organized.
- It addresses an important practical problem: ensuring privacy in treatment effect estimation from sensitive medical data.
- The proposed DP-CATE framework is highly flexible and model-agnostic, compatible with any doubly robust meta-learner and machine learning model.
- The authors provide theoretical guarantees for differential privacy while preserving the double robustness property.

**Weaknesses:**

- The paper does not sufficiently analyze the consistency of the proposed estimators, i.e., whether the estimators remain consistent.
- The presentation of the identification condition (3) is unclear. The authors should clarify the assumptions (e.g., unconfoundedness) under which the optimizer of (3) represents the true conditional average treatment effect function.
- Theorem 1 builds upon the work of Avella-Medina (2021). The authors seek an upper bound on $\zeta$-smooth sensitivity to ensure privacy. However, is this bound tight, and might there be a more optimal bound for $\zeta$-smooth sensitivity?
- Could the authors comment on the inclusivity of the two proposed methods? For example, if one generates a functional query and then uses it to answer finite queries, what would be the potential advantages or disadvantages of this approach?

**Questions:**

See weaknesses.

---

> ### Author Response · Authors · 2024-11-20
> **Response to reviewer qbNM**
>
> Dear reviewer qbNM,
>
> Thank you for your feedback on our manuscript! We took all your comments at heart and improved our paper accordingly. Below, we provide answers to the questions in your review. We **updated our PDF** and highlighted all key changes in **blue color**.
>
>
> - **Consistency of DP-CATE:**
> Thank you for the suggestion. We are more than happy to discuss the consistency of our final private meta-learners in detail. Our proposed DP-CATE framework provides consistent CATE estimators as we will outline in the following. Note that our framework does not alter the estimation procedure itself. As we built upon the consistent doubly robust estimators, the CATE estimator without the Gaussian noise is consistent. Furthermore, observe that the amount of noise added decreases in the sample size $n$. Let $g^P$ denote the private estimator, $g$ the non-private base meta-learner and $\tau$ the true CATE. Then
> $$
> \lVert g^P - \tau \rVert \leq \lVert g^P - g \rVert + \lVert g - \tau \rVert \rightarrow 0, n \rightarrow \infty.
> $$
> Therefore, $g^P$ is a consistent estimator of $\tau$.
>
>     **Action**: We have **added a discussion** on the consistency of our framework in a separate appendix dedicated to this topic (see **Supplement D**)
>
>
>
>
> - **Clarification on causal assumptions:**
> Thank you for highlighting that readers from fields outside of causal inference might not be familiar with the necessary assumptions for identifying causal effects from observational data. In the following, we describe the three assumptions in more detail:
> The estimation of causal quantities, such as the conditional average treatment effect $\tau(x) = \mathbb{E}[Y(1) -Y(0) \mid X = x]$ involves counterfactual quantities $Y(a)$, as only one outcome per individual can be observed. Therefore, identification of causal effects from observational data necessitates the following three assumptions common in the literature (e.g. [1],[2],[3]).
>      1. **Positivity/Overlap:** The treatment assignment is not deterministic. Specifically, there exists a positive probability for each possible combination of features to be assigned to both the treated and the untreated group, i.e., $\exists \ \kappa > 0$ such that $\kappa <\pi(x) < 1-\kappa$ for all $X=x \in \mathcal{X}$.
>      2. **Consistency:** The potential outcome $Y_i(a=k)$ equals the observed factual outcome $Y_i$ when individual $i$ was assigned treatment $A_i=k$.
>     3. **Unconfoundedness:** Conditioned on the observed covariates, the treatment assignment is independent of the potential outcomes, i.e., $Y(0), Y(1) \perp A|X$. Specifically, there are no unobserved variables (confounders) influencing both the treatment assignment and the outcome.
> The assumptions are necessary for consistent causal effect estimation for \emph{all} machine learning models. Then, CATE is identifiable as
> $$ \tau(x) := \mathbb{E}[Y(1) -Y(0) \mid X = x] = \mu(x, 1) - \mu(x, 0), $$
> where $\mu(x,a) = \mathbb{E}[Y \mid X=x, A=a]$.
>
>     **Action:** We **added a new section** where we **explain the underlying assumptions** and give a general theoretical background on CATE estimation (see our **new Supplement A3**). We hope that this makes our paper more accessible to readers with a background in differential privacy but without in-depth knowledge of causal inference.
>
>
> - **Tightness of upper bound on sensitivity:**
> Thank you for raising this question. Although tightness of the upper-bound is beneficial as loose upper bounds can lead to overly perturbed estimates, the task of deriving tight bounds is highly non-trivial. This is a common problem in differentially private estimation methods, which employ bounds on the *global sensitivity* as originally demanded by the DP. As in Avella-Medina (2021), it is not possible to prove tightness of the bounds. We thus leave this task for future research.

---

> ### Author Response · Authors · 2024-11-20
> **Response to reviewer qbNM (continued)**
>
> - **Connection of the two proposed approaches:**
> Thank you for giving us the chance to elaborate on the connection between both approaches, especially their differences and similarities. As correctly noted, the functional approach can as well be employed to privately release a finite a-priori known number of CATE estimates. Therein, the approach coincides with the first approach. Nevertheless, the second approach can also be employed for iteratively querying the underlying function. This highly differentiates the approach from the finite setting. The differences in the two ways of how our functional approach can be used can be best seen by inspecting sampling procedure from the Gaussian process $G$. We discuss the connection between the two approaches (finite-query approach and functional approach) based on the type of query f in the following:
>     1. **Simultaneous finitely many queries:** For querying the function **only once** with a finite amount of queries, *sampling from a Gaussian process* implies sampling from the *prior distribution* of the process. In empirical applications, this means that one samples from a multivariate normal distribution. Therefore, the noise added in the functional approach is similar to the finite-query approach. However, the approaches are *not the same*, as the noise added in the functional approach is correlated, whereas the noise variables in finite-query approach are independent. Still, both approaches guarantee privacy.
>     2. **Iteratively querying the function:** In this setting, *sampling from a Gaussian process* implies sampling from the *posterior distribution* of the process. Specifically, if no query has been made to the private function yet, the finite-query approach proceeds by providing the first private
> 	CATE estimate of query $x_1$. Observe that the privatization of every further
> 	iterative query $x_i$ needs to account for the information leakage through
> 	answering former queries. Thus, sampling from a Gaussian process now
> 	relates to sampling from the posterior distribution. To do so, it is
> 	necessary to keep track and store former queries $x_1,\ldots,x_{i-1}$ and the
> 	privatized outputs. This setting is entirely different from our finite query approach in
> 	which we propose to add Gaussian noise scaled by the gross-error sensitivity.
>
>     **Action:** We added a **new supplement** where we **discuss the connection between both approaches** (see our **new Supplement C**). Furthermore, we provide an alternative algorithm for implementing DP-CATE for functions (see our**new Supplement C.1**).
>
>
>
> [1]	Alicia Curth and Mihaela van der Schaar. Nonparametric estimation of heterogeneous treatment effects: From theory to learning algorithms. In Conference on Artificial Intelligence and Statistics (AISTATS), 2021.
>
> [2]	Stefan Feuerriegel, Dennis Frauen, Valentyn Melnychuk, Jonas Schweisthal, Konstantin Hess, Alicia Curth, Stefan Bauer, Niki Kilbertus, Isaac S. Kohane, and Mihaela van der Schaar. Causal machine learning for predicting treatment outcomes. Nature Medicine, 30(4):958–968, 2024.
>
> [3]	 Donald. B. Rubin. Causal inference using potential outcomes: Design, modeling, decisions. Journal of the American Statistical Association, 2005.

---

> > ### Comment · Reviewer_qbNM · 2024-11-25
> >
> > Thank you for the clarification; my confusion has been resolved.  I will maintain my score with a leaning towards acceptance.

---

### Official Review · Reviewer_Ucae · 2024-10-31

**Soundness:** 4
**Presentation:** 4
**Contribution:** 3
**Rating:** 6
**Confidence:** 3

**Summary:**

This paper presents  a new framework (DP-CATE) for estimating conditional average treatment effects (CATE) under differential privacy while ensuring double robustness (see below my comment on robustness). DP-CATE is broadly applicable to two-stage CATE meta-learners with a Neyman-orthogonal loss. The framework can perform both pointwise estimation of the CATE function and direct functional estimation. The authors provide experimental results on synthetic and real data that demonstrate the effectiveness of DP-CATE.

**Strengths:**

- The paper is exceptionally well-written and easy to follow, providing a clear presentation of complex concepts.
- The authors present a well-founded methodology and provide rigorous mathematical analysis.
- Experimental results on both synthetic and real data help to validate the proposed framework.
- The framework is general and flexible, covering a wide range of learning algorithms without making unrealistic assumptions.
- The discussion in line 314 about the relationship between doubly robust learners and smooth sensitivity offers an interesting insight that could have broader implications.

**Weaknesses:**

- While the authors discuss the novelty of their work on functional data privacy, they underplay previous work in this area. Notably, Hall et al. (2013) provides a mechanism for functional data, which the authors should highlight more explicitly to offer better context for readers.
- The use of “robustness” could be misleading, as it carries different meanings across fields. Clarifying what robustness specifically entails here would be helpful, especially considering existing work on “privacy and robustness.”
- Certain definitions, such as those on line 157, could be recalled for clarity. Additionally, the definition of Y ( ⋅ )    used on line 160 appears to be missing, which may cause confusion.
Overall, while these issues do not significantly detract from the quality of the work, addressing them could improve clarity and reader understanding.

**Questions:**

- How does the proposed approach for functional data compare to the techniques in Hall et al. (2013)? Highlighting these distinctions would provide valuable context for readers.
- While the framework is general, how does its performance compare numerically to similar algorithms (e.g.,Betlei et al. (2017), Guha & Reiter (2024) and Niu et al. (2019)) in settings where those methods are applicable?
- Could the authors provide insights on how functional estimation compares to pointwise estimation in practical scenarios?

---

> ### Author Response · Authors · 2024-11-20
> **Response to reviewer Ucae**
>
> Dear reviewer Ucae,
>
> Thank you for your positive and detailed review of our paper! We took all your comments at heart and improved our paper accordingly. We **updated our PDF** and highlighted all key changes in **blue color**.
>
> **Answer to weaknesses:**
>
> 1. **Relation to Hall (2013):**
> Thank you for this suggestion. We cite Hall (2013) more carefully to offer context to our readers. We further discuss the differences to Hall (2013) as part of the section “Answers to questions” section below.
>
>     **Action:** We referred to Hall (2013) more often to provide more context to our readers. Thereby, we also spell out the differences between Hall (2013) and our work.
>
>
> 2. **Terminology (robustness):**
> Thank you for giving us the chance to clarify the meaning of robustness and improve our terminology. Throughout our paper, we focus **doubly robust** estimators. This type of robustness implies that the estimators remain consistent even when either the outcome or the treatment selection model (i.e., the propensity model) is not correctly specified. The estimators are **insensitive to perturbations** of the nuisance models. We apologize that we also used the term *robustness* in other places to refer to the insensitivity to outliers. Upon reading your comment, we realized that this could be confusing to the reader. => We rewrote the respective texts and improved our terminology. In particular, we clearly distinguish between “double robustness” and *flexibility* or *insensitivity*, where appropriate.
>
>     **Action:** We updated the terminology in our manuscript. Additionally, we included a **new section** where we give **theoretical background on the double robustness property** (**Supplement A3**).
>
> 3. **Recalling definitions:**
> Thank you for highlighting that readers from fields outside of causal inference might not be familiar with certain definitions and notations. We thus added more background materials throughout our manuscript. We hope that this makes our paper more accessible to readers without in-depth knowledge of causal inference.
>
>     **Action:** We have **introduce** important the definitions throughout the manuscript. Additionally, we included a **new section** where we give **theoretical background on CATE estimation** and the underlying assumptions (**Supplement A3**).

---

> > ### Author Response · Authors · 2024-11-20
> > **Response to reviewer Ucae (continued)**
> >
> > **Answer to questions:**
> >
> > - **Comparison to Hall (2013):**
> > Thank you again for highlighting that a more detailed differentiation to Hall (2013) offers better context for the readers. Our work builds upon Hall (2013) in the sense of considering functions in a RKHS. This allows us to employ Corollary 9 and parts of Section 4.3 of Hall (2013) in our derivations. However, **the design of employing a Gaussian kernel ridge regression in the second stage regression and the derivations thereupon are unique to our work** and **thus different from Hall (2013)**. Our contribution lies especially in the transfer from previous related work to the causal inference domain.
> >
> >     **Action:** We cited Hall (2013) and similar works more frequently to provide more context to readers. We also spelled out the differences more clearly.
> >
> >
> > - **Baselines:**
> > Thank you for giving us the chance to show the superiority of our approach to other, more restrictive methods. We highlight again that other DP methods are either not model-agnostic (Niu et al. (2019)), or restricted to RCT data, or restricted to data with binary outcomes (Betlei et al. (2017), Guha & Reiter (2024)). Therefore, the baselines are not applicable to general settings. Nevertheless, we agree with the reviewer that a comparison to Niu et al. (2019) is of interest, as the method is – in principle – applicable to the datasets we employ. We thus **performed new experiments with Niu et al (2019) as a baseline**. We report the results in our updated manuscript (see our **new Supplement E**). We find that the error induced by privatization from DP-EBM (the method in Niu et al) is – by far – worse than the error induced by our DP-CATE. We explain this is due to the need for privatizing both stages in the DP-EBM framework while ours is end-to-end. In sum, **this confirms that our method is superior** to the DB-EBM from Niu et al (2019).
> >
> >     **Action:** We present an evaluation and **comparison to the DP-EBM** method presented in Niu et al. (2019) in our **new Supplement E**.

---

> ### Author Response · Authors · 2024-11-20
> **Response to reviewer Ucae (continued 2)**
>
> - **Comparison of both approaches (finite and functional DP-CATE):** Thank you for giving us the opportunity to explain the differences between the two proposed approaches in practical scenarios. If one only wants to release private CATE estimates *once*, both approaches are applicable. Nevertheless, the second approach called “functional approach” can also be employed for iteratively querying the function, which is especially of interest to medical practitioners aiming to assess the treatment effect of a drug for various patients with different characteristics. Put simply, when companies want to release a decision support system to guide treatment decisions of **individual patients**. Such treatment decisions are made based on the entire CATE **model**, then the “functional” approach is preferred. In contrast, the first approach (called “finite-query approach”) is preferred whenever only a few CATE **values** should be released. This is relevant for researchers (or practitioners) who may want to share the treatment effectiveness for a certain number of **subgroups** (but not for individual patients).
>
>     The functional approach requires sampling from a Gaussian process. Depending on wheter one aims to report finitely many querie once through this approach or iteratively query the function, the sampling procedure from the Gaussian process $G$ differs. We highlight the differences of the type of queries in the following:
>
>     1. **Simultaneous finitely many queries:** For querying the function **only once** with a finite amount of queries, *sampling from a Gaussian process* implies sampling from the *prior distribution* of the process. In empirical applications, this means that one samples from a multivariate normal distribution. Therefore, the noise added in the functional approach is similar to the finite-query approach. However, the approaches are *not the same*, as the noise added in the functional approach is correlated, whereas the noise variables in finite-query approach are independent. Still, both approaches guarantee privacy.
>     2. **Iteratively querying the function:** In this setting, *sampling from a Gaussian process* implies sampling from the *posterior distribution* of the process. Specifically, if no query has been made to the private function yet, the finite-query approach proceeds by providing the first private CATE estimate of query $x_1$. Observe that the privatization of every further iterative query $x_i$ needs to account for the information leakage through answering former queries. Thus, sampling from a Gaussian process now relates to sampling from the posterior distribution. To do so, it is necessary to keep track and store former queries $x_1,\ldots,x_{i-1}$ and the privatized outputs. This setting is entirely different from our finite setting approach in which we propose to add Gaussian noise scaled by the gross-error sensitivity.
>
>     **Action:** We included a **new supplement** where we **discuss the connections** between the two approaches (see our **new Supplement C**). Furthermore, we provide an alternative algorithm for implementing DP-CATE for functions (see our new **Supplement C.1**).

---

> > ### Comment · Reviewer_Ucae · 2024-11-22
> >
> > Thank you for your detailed response and the significant clarifications provided. I appreciate the improved discussion on the differences with Hall (2013), the new experiments with additional baselines, and the comprehensive explanation of the two proposed approaches. These revisions have greatly enhanced my confidence in the paper’s contributions and its theoretical foundations. While I will maintain my rating of 6, I have increased my confidence score based on the authors’ clarifications and updates.

---

### Official Review · Reviewer_smF6 · 2024-11-02

**Soundness:** 3
**Presentation:** 4
**Contribution:** 2
**Rating:** 8
**Confidence:** 3

**Summary:**

This paper studies the problem of causal inference with sensitive observational data with DP, motivated by medical applications.  specifically, the authors study estimtaing the CATE function (conditional average treatment effect), which as the name implies, quantifies the effect of a treatment as a function of some covariate.  The authors propose a simple output perturbation mechanism to estimate the CATE with DP.

**Strengths:**

* The paper is technically sound and statistically rigorous.
* The problem studied is novel and of practical interest.
* The paper is clearly written and nicely polished.

**Weaknesses:**

* The paper uses some jargon that may not be familiar to the reader (e.g., doubly-robust).
* There are no comparisons to baselines.  While the problem studied is new there are no published baselines on this approach, there are some simple baselines you could compare against.
>* Here's one: in 1D case you study in e.g., Fig 1, 6 just compute the CATE function non-parametrically.  That is, compute COUNT(Y=1, a <= X <= b) and COUNT(Y=0, a <= X <= b) for a variety of intervals in the domain, from which you can estimate CATE easily.
>* To handle higher dimensional case, you could compute those counts for each covariate and then make some kind of conditional independence assumption.  In the p=2 case you consider in experiments, you could also just directly compute the full histogram and it should be pretty doable.   I would assume there are both qualitative and quantitative advantages to your approach, but I think it would be good to demonstrate that explicitly.
>* From the idea above, it seems this can be framed as a marginal-preservation problem, a problem that many synthetic data algorithms are pretty good at (e.g., PrivBayes).  That could be another baseline.

**Questions:**

* In the problem statement, do you have any constraints on the domain of X?  In practice is it usually a small domain (e.g., one feature) or large domain (many features)?
*  In the finitely many queries setting, can you be more precise about the typical characteristics of the setting?  How many queries is typical, and what do those queries look like?
>* If the number of queries is small (e.g., quantify CATE for ages 0-20, 20-30, 30-40, ...,) then the problem is trivial.
>* I think it would be good to give a motivating example to make the abstract problem formulation you have a bit more grounded.

---

> ### Author Response · Authors · 2024-11-20
> **Response to reviewer smF6**
>
> Dear reviewer smF6,
>
> Thank you for your positive evaluation of our paper! We took all your comments at heart and improved our paper accordingly. We **updated our PDF** and highlighted all key changes in **blue color**.
>
> **Answer to weaknesses:**
>
> 1. **Jargon:**
> Thank you for giving us the chance to improve our terminology. Throughout the paper, we consider so-called **doubly robust** estimators for causal inference. Double robustness implies that the estimators remain consistent even when either the outcome or the treatment selection model (i.e., the propensity model) is not correctly specified. The estimators are thus **insensitive to perturbations** of the nuisance models, which is a great practical benefit.
>
>     **Action:** We simplified the terminology in our manuscript. Additionally, we included a **new section** where we give **theoretical background on the double robustness property** (**Supplement A3**). We hope that this makes our paper more accessible to readers with a background in differential privacy but without in-depth knowledge of causal inference.
>
> 2. **Comparison to baselines:**
> Thank you for giving us the chance to show the superiority of our approach to other more restrictive methods. We highlight again that other DP methods are either not model-agnostic (Niu et al. (2019)), or restricted to RCT data or restricted to data with binary outcomes (Betlei et al. (2017), Guha & Reiter (2024)). Therefore, _the baselines are not applicable to general settings as in our paper_. In other words, powerful baselines with theoretical DP guarantees are missing.
>
>
>     We therefore appreciate the reviewer’s suggestion about a very naive baseline based on k-anonymization. As we show in the following, such naive baseline has clear limitations in both theory and in our experiments. First, such k-anonymization essentially performs a data aggregation as part of the privatization technique, yet this can **break the consistency assumption** necessary for causal identifiability. Hence, this can lead to **biased** results. Further, it does not offer formal privacy guarantees as our method. Nevertheless, we agree with the reviewer that a comparison to such a more naive privacy method is of interest. Hence, we implemented the method and performed **new experiments**. We observe that our method provides superior CATE estimates (while further offering theoretical guarantees to ensure DP) for almost all privacy budgets. Hence, our new experiments confirm again the effectiveness of our proposed method.
>
>
>     **Action:** We added a new experiment where we compare our method to a naive baseline based on k-anonymization, finding that our method is clearly superior (see our **new Supplement E**).
>
>
> **Answer to questions:**
>
> 1. **Domain of X:**
> The only constraint on the domain of X is that it is bounded, which is very reasonable in practice. We do not make any further assumptions. In practice, the domain of X can vary from only very few features to high-dimensional settings. We aimed to cover both settings in our synthetic experiments on datasets with 2 and 30 confounders, respectively.
>
>     **Action:** We stated the assumptions on the domains of the variables more clearly in our manuscript.
>
>
> 2. **Characteristics of setting 1:** Thank you for raising this question. We agree that further motivation would greatly improve comprehension, and we are thus happy to offer additional clarifications. In practice, the queries in this setting can range from a very small number, such as different age groups, as suggested by the reviewer, to very large numbers of CATE estimates, such as for various combinations of patient characteristics. In our experiments, the number of queries varied from 300 queries
> (synthetic experiments) to 1312 queries (MIMIC) and 2659 queries (TCGA).
>
>     **Action:** We included more elaborations to explain our experimental setup.

---

> ### Author Response · Authors · 2024-11-25
> **Follow-up on rebuttal**
>
> Dear reviewer smF6,
>
> We hope we sufficiently addressed all your concerns in our rebuttal. If you have any more questions or concerns, we are happy to answer them as soon as possible. Please let us know if this is the case. Otherwise, we would highly appreciate it if you could raise your evaluation score for our manuscript.
>
> Best regards,
> The authors

---

> > ### Comment · Reviewer_smF6 · 2024-11-26
> >
> > Thanks for the detailed response and for incorporating the feedback.  I think this paper is above the bar for ICLR, and have updated my score accordingly in the spirit of taking a firm stance.

---

### Official Review · Reviewer_ukCv · 2024-11-03

**Soundness:** 3
**Presentation:** 2
**Contribution:** 2
**Rating:** 6
**Confidence:** 4

**Summary:**

This study proposes two new methods to address the problem of computing CATE estimates under differential privacy. Leveraging the often used assumption of number of queries being known a priori the authors develop a method that uses influence functions as an upper bound to the smooth sensitivity quantity. This is used in a traditional output perturbation algorithm of the CATE estimate to calibrate the Gaussian noise. The second method is for releasing the CATE function in its entirety under differential privacy. This is a much more difficult problem. The authors develop an algorithm that guarantees DP by using a calibrated Gaussian process to modify the output of the original CATE algorithm. They develop an algorithm for determining how to calibrate this process leveraging theory about RKHSs and Gaussian kernel regression. The efficacy of these algorithms are demonstrated on synthetic datasets where access to the ground truth CATE is available and observational medical datasets.

**Strengths:**

- By and large this is a well-written paper and it was easy to understand what both algorithms were doing at a high-level.
- The methods provide more flexibility than prior work in supporting all types of ML models
- The separation between knowing the number of queries a priori and not is well taken as it allows them to build a stronger estimator in the fixed query setting
- The experiments show some promise that the estimator is relatively close to the ground truth in the synthetic experiments

**Weaknesses:**

- The algorithm for releasing the CATE function assumes knowledge of the Lipschitz constant which unless I am mistaken seems like an unrealistic assumption (note I am not concerned with assuming Lipschitzness of the loss)
- More details about the experiments are needed to help understand them. Right now the details are quite sparse so it’s difficult to contextualize them in each of the challenges described. I leave my questions related to this for the Questions section of the review.
- Unless I am misunderstanding Figure 6 and 7, it seems like th CATE is constant along the ages / covariates for each task? If so, this is not as compelling for the success of this method. I think like the synthetic dataset, an observational task should be chosen where the CATE differs based on the covariate I will be adjusting. It’s important to understand how well the DP estimator can capture the variation in CATE as the condition covariate changes.

**Questions:**

1. How many queries were done for Dataset 1 (Figure 3 and 4)?
2. Do the authors have a sense for how tight they think the upper bound for smooth sensitivity is using the gross error sensitivity? Is there room to make it tighter with more assumptions?
3. In Figure 4, it seems like DP-CATE consistently underestimates the CATE at the 0.01 privacy budget. Is there any intuition and/or concrete results on whether this method tends to underestimate or overestimate?
4. What does y-axis represent in Figure 6 and 7? Is it the actual CATE or is it the error?
5. How was the Lipschitz constant computed for the empirical results?

---

> ### Author Response · Authors · 2024-11-20
> **Response to reviewer ukCv**
>
> Dear Reviewer ukCV,
>
>
> Thank you for your detailed feedback on our manuscript. We took all your comments at heart and improved our manuscript accordingly. We **updated our PDF** and highlighted all key changes in **blue color**.
>
> **Response to weaknesses:**
>
> - **Knowledge of the Lipschitz constant:**
> Thank you for raising this question. We agree that it might not directly become clear why the assumption of the knowledge of the Lipschitz constant is not restrictive.
> First, employing the lipschitz constant of the loss for post-hoc processing is very common across many fields (e.g.[1],[2],[3],[4]). Second, for many losses, the Lipschitz constant is data-independent and directly computable from the loss function. For example, for the L1 loss, the Lipschitz constant L equals 1; for the Huber loss L equals the loss parameter $\delta$, or for the truncated L2 loss, the constant equals the gradient at the truncation value.
> **Action:** We added a discussion of the Lipschitz constant to the updated version of our manuscript.
>
> - **Details on the experiments:**
> Thank you! We apologize for not stating sufficient details on how we conducted our experiments. We greatly extended our experimental details. We also respond below to your specific questions and how we improved our paper as a result (see our “Response to question”).
> **Action:** We added more details on the experiments to our manuscript in the respective parts in the main paper and the corresponding supplement.
>
> - **Real-world dataset with heterogeneous treatment effect:**
> Thank you for giving us the chance to show the applicability of DP-CATE to medical datasets with heterogeneous treatment effects. Upon reading your question, we realized that we chose a clinical example where the effect is constant (i.e., the effect of ventilation on clinical outcomes is known to have little variation across patients of different ages) and therefore comes without interesting clinical interpretation. As a result, we have opted for a different example with direct clinical interpretation. Specifically, we now present a **new experiment** in which we estimate the effect of ventilation on red blood cell count conditioned on hematocrit (see our **new Fig. 6**). Here, we should expect a nice positive relationship as stipulated by domain knowledge in medicine. Aligned with this, we indeed with an increasing effect across different hematocrit values.
> **Action:** We reworked our experiment for the MIMIC dataset to show a more meaningful example with clinical interpretation (see our **new Figure 6**).

---

> ### Author Response · Authors · 2024-11-20
> **Response to reviewer ukCv (continued)**
>
> **Answer to Questions:**
>
> - **Queries for setting 1 (Fig. 3+4):** Thank you for spotting this. We apologize that we did not state the number of queries we evaluated our framework on. In the synthetic experiments (Fig. 3+4), we evaluated DP-CATE on 300 queries. The other experiments were evaluated on 1312 queries (MIMIC) and 2659 queries (TCGA).
> **Action:** We added this to the experimental results.
>
> - **Tightness of the sensitivity bound by the gross-error sensitivity:**
> Thank you for raising this question. Although tightness of the upper-bound is beneficial as loose upper bounds can lead to overly perturbed estimates, the task of deriving tight bounds is highly non-trivial. This is a common problem in differentially private estimation methods, which typically employ bounds on the *global sensitivity* as originally demanded by the DP. As in Avella-Medina (2021), it is unfortunately not possible to prove the tightness of the bounds. We thus leave this task for future research.
>
> - **Question on functional DP-CATE in Fig.4:**
> Thank you for this interesting question. The reviewer is right that in the shown plot DP-CATE underestimates the true CATE. However, there is no consistent under- or overestimation of the method for certain settings. Rather, this behavior is merely a result of the privatization approach in the functional setting 2. In contrast, in setting 1, we sample $d$ independent random normal variables, while, in setting 2, we sample from a Gaussian process through sampling a multivariate normal variable with the corresponding covariance. Thus, the added noise is correlated. This might result in what appears to be consistent under- or overestimation of the target.
> **Action:** We included a **new supplement** where we **discuss the differences and similarities** of the two approaches and give more intuition on the behavior of setting 2 (**Supplement C**).
>
> - **Y-axis in Fig. 6+7:**
> In Figure 6+7, the y-axis represents the CATE, not the estimation error.
> **Action:** We stated the presented quantities more precisely in the updated version of our manuscript.
>
> - **Computation of the Lipschitz constant for the empirical evaluation:**
> Thank you for noting that we did not specify how we calculated the Lipschitz constant in our empirical evaluation. In our settings, we employ the L2-loss on a bounded domain.
> Therefore, although the L2-loss itself is not Lipschitz, we can calculate L numerically as the gradient at the upper bound of the loss.
> **Action:** We stated the computation of the Lipschitz constant in our empirical evaluation more precisely in the updated version of our manuscript.

---

> ### Author Response · Authors · 2024-11-25
> **Follow-up on rebuttal**
>
> Dear reviewer ukCv,
>
> We hope we sufficiently addressed all your concerns in our rebuttal. If you have any more questions or concerns, we are happy to answer them as soon as possible. Please let us know if this is the case. Otherwise, we would highly appreciate it if you could raise your evaluation score for our manuscript.
>
> Best regards,
> The authors

---

> > ### Comment · Reviewer_ukCv · 2024-12-01
> > **Apologies for delay in response.**
> >
> > Sincere apologies for the delay in response. The authors have sufficiently addressed my concerns. I will update my score accordingly.

---

### Author Response · Authors · 2024-11-20
**Response to all reviewers**

Thank you very much for the constructive evaluation of our paper and your helpful comments! We addressed all of them in the comments below.

Our main improvements are the following:

- We added **new theoretical results** where we show the consistency of our framework (see our new **Supplement D**). We further discuss the similarities and differences between our two proposed settings (see our new **Supplement C**). Furthermore, we provide an alternative algorithm for implementing DP-CATE for functions (see our new **Supplement C.1**).

- We added **new empirical results** to demonstrate the superiority of our framework over further baselines (see our new **Supplement E**). Again, our proposed framework performs best.


- We added **new empirical results** on the MIMIC dataset to show the applicability of our framework to data with heterogeneous treatment effects (see **Section 5**). Again, our proposed framework is highly effective.

- We added a **theoretical background** on CATE estimation in which we detail the causal assumption and give more background on the meta-learners (see our new **Supplement A.3**). We hope that this makes our paper more accessible to readers with a background in differential privacy but without in-depth knowledge of causal inference.

We incorporated all changes into the **updated version of our paper**. Therein, we highlight all key changes in **blue color**. Given these improvements, we are confident that our paper will be a valuable contribution to the literature on differential privacy in causal effect estimation and a good fit for ICLR 2025.

---

### Meta-Review · Area_Chair_JXQy · 2024-12-17

**Metareview:**

The paper presents two methods for differentially private (DP) causal inference by conditional average treatment effect estimation.

The reviewers agree that the paper makes important advances in this problem and is well-presented.

The reviewers do not identify significant weaknesses that would prevent publication and all reviewers recommend acceptance.

Therefore the paper should be accepted.

**Additional Comments On Reviewer Discussion:**

All reviewers responded to author response, noting that it addressed their concerns and recommended acceptance.

---

### Decision · Program_Chairs · 2025-01-22

Accept (Poster)